



# Lagrangian matches between observations from aircraft, lidar and radar in an orographic warm conveyor belt

Maxi Boettcher[1], Andreas Schäfler[2], Michael Sprenger[1], Harald Sodemann[3], Stefan Kaufmann[2], Christiane Voigt[2,4], Hans Schlager[2], Donato Summa[5], Paolo Di Girolamo[6], Daniele Nerini[7], Urs Germann[7], and Heini Wernli[1]

[1]Institute for Atmospheric and Climate Science, ETH Zurich, Switzerland
[2]Institute of Atmospheric Physics, German Aerospace Center, Oberpfaffenhofen, Germany
[3]University of Bergen, Norway
[4]Institute for Atmospheric Physics, Johannes Gutenberg-University Mainz, Germany
[5]Consiglio Nazionale delle Ricerche,Tito Scalo, Italy
[6]Scuola di Ingegneria, Universita degli Studi della Basilicata, Potenza, Italy
[7]MeteoSwiss, Switzerland

**Correspondence:** Maxi Boettcher, Institute for Atmospheric and Climate Science, ETH Zurich, Universitaetstrasse 16, 8092 Zurich, Switzerland (maxi.boettcher@env.ethz.ch)

**Abstract.** Warm conveyor belts (WCBs) are important airstreams in extratropical cyclones, often leading to the formation of intense precipitation and the amplification of upper-level ridges. This study presents a case study that involves aircraft, lidar and radar observations in a WCB ascending from western Europe towards the Baltic Sea during the field experiments HyMeX and T-NAWDEX-Falcon in October 2012.

Trajectories were used to link different observations along the WCB, that is to establish so-called Lagrangian matches between observations. To this aim, wind fields of the ECMWF ensemble data assimilation system were used, which allowed for a probabilistic quantification of the WCB occurrence and the Lagrangian matches. Despite severe air traffic limitations for performing research flights over Europe, the DLR Falcon successfully sampled WCB air masses during different phases of the WCB ascent. The WCB trajectories revealed measurements in two distinct WCB branches: one branch ascended from the east-
ern North Atlantic over southwestern France, while the other had its inflow in the western Mediterranean. Both branches passed across the Alps, and for both branches, Lagrangian matches coincidentally occurred between lidar water vapour measurements in the inflow of the WCB south of the Alps, radar measurements during the ascent at the Alps, and in situ aircraft measurements by Falcon in the WCB outflow north of the Alps. An airborne release experiment with an inert tracer could confirm the long pathway of the WCB from the inflow in the Mediterranean boundary layer to the outflow in the upper troposphere near the
Baltic Sea several hours later.

The comparison of observations and ensemble analyses reveals a moist bias in the analyses in parts of the WCB inflow but a good agreement of cloud water species in the WCB during ascent. In between these two observations, a precipitation radar measured strongly precipitating WCB air located directly above the melting layer while ascending at the southern slopes of the Alps. The trajectories illustrate the complexity of a continental and orographically influenced WCB, which leads to (i) WCB
moisture sources from both the Atlantic and Mediterranean, (ii) different pathways of WCB ascent affected by orography, and





(iii) locally steep WCB ascent with high radar reflectivity values that might result in enhanced precipitation where the WCB flows over the Alps. The linkage of observational data by ensemble-based WCB trajectory calculations and confirmed by an inert tracer, and the model evaluation using the multi-platform observations are the central elements of this study and reveal important aspects of orographically modified WCBs.

## 1 Introduction

The warm conveyor belt (WCB) is one of the three coherent airstreams described by, e.g. Carlson (1980) and Browning (1990) in the so-called conveyor belt model of extratropical cyclones. The WCB is the airstream that ascends the most, rising from the boundary layer in the cyclone's warm sector to the upper troposphere downstream of the cyclone within about one day (e.g. Browning, 1986; Wernli, 1997). Climatologically, WCBs are most frequent in the winter season and they preferentially occur

in the main oceanic storm track regions in both hemispheres (Madonna et al., 2014b). WCBs transport heat, moisture, and atmospheric pollutants typically poleward (e.g. Methven et al., 2006; Sinclair et al., 2008). They are, due to the intense formation of clouds in the ascending airstream, among the strongest precipitating weather systems in the mid-latitudes (Eckhardt et al., 2004; Pfahl et al., 2014; Catto et al., 2015). Early studies about WCBs focused on clouds and precipitation structure (Browning, 1971; Harrold, 1973). They mainly relied on cyclone-relative isentropic flow analysis, satellite images, and radar observations

to identify the WCB, which appear as hammer-shaped cloud bands that are characteristic for extratropical cyclones (e.g. Carlson, 1980; Browning, 1986; Young et al., 1987). The recent study by Joos (2019) investigated the substantial radiative forcing exerted by the extended cloud shields associated with WCBs. Observational studies (e.g. Muhlbauer et al., 2014; Krämer et al., 2016; Luebke et al., 2016; Voigt et al., 2017) revealed the differing microphysical properties of WCB cirrus compared to other types of cirrus and highlighted their potential importance for radiation and climate.

This brief summary of general characteristics of WCBs emphasises their importance for mid-latitude weather including heavy precipitation (e.g. Pfahl et al., 2014). For numerical weather prediction (NWP), WCBs still constitute one of the most challenging weather elements (Rodwell et al., 2018) due to their troposphere-deep motion accompanied by cloud-thermodynamical processes from typically $+20$ to $-50°$C. The presented study investigates a special case of an orographically influenced WCB over Central Europe in autumn 2012 using a sequence of measurements taken during the field campaigns

HyMeX and T-NAWDEX-Falcon. This sequence of measurements that are physically related to each other by air parcel trajectories is referred to as *Lagrangian matches* (Methven et al., 2006). To the best of our knowledge, this is the first study that describes Lagrangian matches between humidity measurements in a WCB. To provide direct experimental evidence for the long-range transport by this WCB, an unique airborne tracer release experiment was performed as part of T-NAWDEX-Falcon. The measurements mainly portray the evolution of moisture along the WCB ascent, which is, on the one hand, one the most

important characteristics of the WCB, and on the other hand, still seen as an obstacle for accurate weather forecasts (e.g. Joos and Forbes, 2016; Rodwell et al., 2018). The following paragraphs will further discuss (i) why moisture in the WCB inflow is critical for the WCB ascent, (ii) why the evolution of moisture and related diabatic processes along the WCB ascent is important, and (iii) why observations of moisture in WCBs are a valuable supplement to model studies.





Moisture and moist processes, which we address by our measurements, are essential characteristics of the WCB. The cloud
formation in the WCB can modify the dynamics of the governing weather system, and the associated latent heat release is
essential to achieve the troposphere-deep moist isentropic ascent of WCBs (Wernli, 1997; Schemm et al., 2013). In addition,
latent heat release feeds back on the flow dynamics by modifying potential vorticity (PV) (e.g. Hoskins et al., 1985; Wernli and
Davies,1997). To first order, latent heat release due to condensation of water vapour in the ascending WCB leads to positive
and negative PV changes below and above the level of maximum condensation, respectively (Wernli and Davies, 1997). For
instance, the diabatically produced low-level positive PV anomaly during WCB ascent can have an impact on the intensity
of the surface cyclone (Binder et al., 2016; Attinger et al., 2019), while the negative PV anomaly in the WCB outflow can
interact with the upper-tropospheric wave guide, leading to wave amplification, Rossby wave breaking, and the formation of
PV streamers (e.g. Stoelinga, 1996; Massacand et al., 2001; Grams et al., 2011; Davies and Didone, 2013; Madonna et al.,
2014a).

The simple concept of a diabatically produced PV dipole during the WCB ascent introduced by Wernli and Davies (1997)
has been refined in the last decade in at least two important ways. First, it became clear that the microphysical processes
within WCBs are more complex, and second, convection was found to potentially occur embedded within the large-scale
cloud band of the WCB. Considering microphysics, Joos and Wernli (2012) were the first to study the detailed cloud structure
of a WCB in a regional numerical weather prediction (NWP) model. The water species in a WCB evolve along the ascent
from low-level water vapour, condensation and the formation of liquid clouds, freezing and the formation of mixed-phase
clouds, up to cirrus near the tropopause. Phase transitions also occur beneath the strongly ascending WCB air parcels due
sedimenting and potentially sublimating of melting hydrometeors (e.g. Crezee et al., 2017). Governed by such a variety of
microphysical processes, this leads to a complex pattern of latent heating in the WCB and regions with latent cooling beneath.
The modelling study by Joos and Forbes (2016) showed that the vertical cloud structure of a WCB is affected by seemingly
small changes in the implementation of the microphysics, with direct implications for the detailed shape and amplitude of
the upper-tropospheric ridge influenced by the WCB outflow. More recently, Binder et al. (2020) systematically compared the
vertical structure of WCB-related clouds in the latest ECMWF reanalysis dataset ERA5 (Hersbach et al., 2020) with profiles
derived from satellite-based lidar and radar observations and concluded that the model reproduces the overall cloud structure
quite well, but underestimates ice and snow water content in the mixed-phase layer in WCBs above the melting layer. Such
weaknesses in models might arise from various assumptions made in microphysical parameterizations that account for the
sub-grid scale nature of the cloud processes and often lead to uncertainties in NWP (Illingworth et al., 2007; Rodwell et al.,
2018). The phenomenon of WCB-embedded convection was already pointed out in very early observational WCB studies (e.g.
Browning, 1971), then brought up again by Neiman et al. (1993). Recently, Flaounas et al. (2016) and Oertel et al. (2019)
confirmed the presence of convection embedded in WCBs with radar observations. Oertel et al. (2020), and Oertel et al. (in
preparation) showed that WCB-embedded convection can result in locally enhanced precipitation and that diabatic heating by
convection can influence the upper-level jet by the formation of PV dipole bands.

This brief summary reveals the importance of WCBs for understanding precipitation in and the dynamics of extratropical
cyclones, but it also indicates the complexity of their ascent behaviour and the involved microphysical processes. It is conceiv-





able that the detailed properties of a WCB strongly depend on the humidity in its inflow. For numerical weather prediction,
the correct representation of humidity particularly in strongly cloudy situations as in WCBs is still one of the main challenges
(Rodwell et al., 2018). However, most of the climatological investigations (e.g. Eckhardt et al., 2004; Madonna et al., 2014b)
and case studies of WCBs (e.g. Joos and Wernli, 2012; Martínez-Alvarado et al., 2016) were based on reanalyses or model
simulations only. Comparatively few studies used observations to evaluate the humidity in WCBs. Schäfler et al. (2011) inves-
tigated airborne lidar measurements of humidity in the low-level inflow of a WCB over Spain and found an overestimation of
humidity in ECWMF analyses. A similar result was found by Schäfler and Harnisch (2015) for the inflow of a marine WCB
over the eastern North Pacific. Their sensitivity experiments with the ECMWF forecast model revealed that with the corrected
low-level moisture in the initial conditions, the WCB outflow would have occurred at a lower potential temperature level and
produced a less developed upper-level ridge. The humidity in the WCB inflow, however, is not only determined by boundary
layer ventilation (Boutle et al., 2011; Pfahl et al., 2014). It can also be affected by a recycling of moisture within the WCB,
which occurs, e.g. when raindrops from an elevated layer of the WCB fall into a sub-saturated lower layer of the WCB inflow
where they evaporate (Crezee et al., 2017; Attinger et al., 2019; Spreitzer, 2020).

Most observational studies of WCBs so far, e.g. based on surface radar measurements (Browning, 1971) or more recently on
aircraft and satellite data (Oertel et al., 2019; Binder et al., 2020), provide limited information about the variability of the WCB
characteristics along the ascent, which typically covers spatial and temporal dimensions of $> 1000\,\mathrm{km}$ and 1 day, respectively.
To better address the Lagrangian nature of these airstreams, a few pioneering field campaign studies attempted to follow the
pathway of a WCB with an aircraft and to measure the same WCB air parcels multiple times (e.g. Stohl et al., 2003; Methven
et al., 2006). In principle, such Lagrangian matches enable investigating the material evolution of thermodynamic variables
along a WCB. A major challenge of such experiments is the fact that the planning of Lagrangian matches with aircraft must
rely on air parcel trajectories using forecast wind fields, which are inherently uncertain. To cope with this uncertainty, the
planning of Lagrangian matches is best done with data from ensemble forecasts (Schäfler et al., 2014; Schäfler et al., 2018). An
interesting observational approach to identify Lagrangian matches is the use of a physical tracer that is measured at consecutive
times to experimentally corroborate the pathway of air parcels. An experiment in 2004 described in Methven et al. (2006) aimed
at realising Lagrangian matches between airborne measurements in the free troposphere to study intercontinental transport of
pollutants. One case of the campaign involved a WCB, for which they used the natural occurrence of their physical tracer to
mark air parcels. The approach in this study is, for the first time, to investigate transport along a WCB by the release and
re-sampling of a synthetic inert tracer (Ren et al., 2015). For completeness, we briefly note that Lagrangian matches have
also been applied in research on stratospheric chemistry, for instance, by Rex et al. (1998) to infer ozone loss rates in the
Arctic stratosphere from ozonsonde measurements, and by Fueglistaler et al. (2002) to study the Lagrangian evolution of polar
stratospheric clouds from consecutive airborne lidar observations.

Since the majority of the WCBs occur over ocean (e.g. Madonna et al., 2014b), not many studies exist about continental
WCBs and the potential effects when they rise over mountains. In their climatological study, Pfahl et al. (2014) found precip-
itation maxima in regions with orographically enhanced WCB ascent. For the European area, a hotspot for WCB ascent and
increased precipitation is found along the Mediterranean side of the Alps. There, over 10% of total precipitation is related to



WCBs and they contribute up to 60% to extreme precipitation events (Pfahl et al., 2014). In a case study, Buzzi et al. (1998) investigated the flood in the Piemont region south of the Alps in November 1994, which was associated with a WCB. Using numerical experiments with flattened orography, they could show that the WCB-related precipitation was distinctly enhanced by orography. But so far, no study investigated how the pathway of WCB trajectories is affected by the interaction with orography.

In October 2012, a team from the German Aerospace Center (DLR) in Oberpfaffenhofen and ETH Zurich organised a small aircraft research campaign, devoted to obtaining in situ measurements of moisture and thermodynamic parameters in different phases of WCBs over Europe (Schäfler et al., 2014). This campaign was termed T-NAWDEX-Falcon and occurred in parallel to the comprehensive HyMeX Special Observation Period 1 (Ducrocq et al., 2014). The WCB presented in this study occurred during the campaign's IOP2, and it ascended from the western Mediterranean across the Alps towards the Baltic Sea. The WCB was successfully sampled by two Falcon flights. The analysis of the airborne observations benefits from additional surface lidar observations made in the framework of HyMeX and from operational radar observations by MeteoSwiss.

In addition, a tracer experiment with the physical release of a passive tracer by a small aircraft in the inflow of the WCB was conducted as part of the T-NAWDEX-Falcon IOP2. The main objective was to "label" WCB air in the inflow and then later "catch" the same air during its ascent by the Falcon aircraft. The use of a passive tracer enables, in principle, a validation whether the air observed at a later time by the Falcon actually had its origin in the labelled WCB inflow region. However, such an experiment can only be successful: (i) if a tracer gas is released that otherwise does not exist in the atmosphere or has a very low atmospheric background concentration like PFCs, such that any observation of the tracer can be uniquely associated with the release experiment; (ii) if a suitable and highly sensitive airborne sampling and analysis technique is available, which allows measuring potentially low concentrations after long-range transport and dilution; (iii) if the forecast is exact enough to reproduce a fairly realistic picture of the actual flow situation that serves as the basis for the flight planning; and (iv) if the flight planning method is appropriate such that the tracer is indeed released in the WCB inflow and that the subsequent Falcon flights targeting the ascending WCB air have a chance of sampling the tracer. In this study we report to what degree we were successful with this challenging experiment.

As shown in detail below, the campaign successfully collected valuable observations in the WCB and benefited from overall good ECMWF forecasts and from the sophisticated multi-stage flight planning procedure (Schäfler et al., 2014). This set of observations will enable us to address the following specific questions related to this orographic WCB case study:

1. How can ensemble analyses be used to provide probabilistic information about the location of WCBs and about Lagrangian matches between observations?

2. How well does the humidity and cloud structure of the WCB in the ECMWF analyses agree with observations from aircraft, lidar, and radar?

3. How closely does the transport in WCBs as given by trajectory calculations with ECMWF analyses correspond to the dispersion of an actually emitted passive tracer?

4. How does the orography of the Alps modify the characteristics of the investigated Central European WCB?





5. What is the overall novel insight about WCBs that can be obtained from such a multi-faceted and observation-based case study?

The manuscript will continue with introducing the measurements and the analysis products, as well as the procedure to identify
Lagrangian matches in section 2. The synoptic situation is described in section 3 before first the results from airborne observations are discussed in subsection 4.1. Lagrangian matches with further observations are described in subsection 4.2, and the tracer experiment in subsection 4.3. Conclusions, including (partial) answers to the questions posed above, are given in section 5.

## 2 Data and Method

This study uses observations from in situ aircraft measurements, from ground-based lidar and radar, respectively, and from a tracer release experiment. The observational datasets are briefly introduced in the following subsections. Then a special diagnostic method, based on ECMWF ensemble analysis data, is introduced, that serves to identify WCBs and Lagrangian matches between measurements.

### 2.1 T-NAWDEX-Falcon humidity measurements

For the T-NAWDEX-Falcon campaign in October 2012 the German DLR Falcon aircraft was operated from the base at DLR in Oberpfaffenhofen near Munich (Schäfler et al., 2014). The Falcon flights on 15 October 2012 were conducted from 7:34 to 10:52 UTC for flight IOP2b and from 13:04 to 16:03 UTC for flight IOP2c. In this study in situ measurements of specific humidity are used that were taken from the basic instrumentation of the aircraft and from the tuneable diode laser system WARAN (Kaufmann et al., 2014; Kaufmann et al., 2018). The Falcon basic humidity measurements for water vapour result
from a composite of three instruments, where emphasis is placed on the lyman-alpha absorption instrument[1]. For flight IOP2b, WARAN was installed with a forward facing inlet so that it collected water of all phases and hence resulted in observations of total water. For flight IOP2c, the inlet was reversed such that no total water measurements were possible during this flight (Voigt et al., 2014; Kaufmann et al., 2016). For IOP2b, the cloud water content was calculated assuming saturation with respect to liquid water for $T > -30°C$ and ice for $T \leq -30°C$, respectively. Particle enhancement at the inlet is corrected assuming
a mean particle radius of $20\,\mu m$ (Krämer and Afchine, 2004). The sampling efficiency of much larger particles like raindrops or snow cannot be quantified directly and thus imposes an additional uncertainty on the measured cloud water content. The two-second in situ measurements of WARAN are smoothed with a 5 min running mean, which corresponds to about the time the aircraft needs to intersect a horizontal grid box of $0.5°$(about $50\,km$) in the model. The same is applied to the one-second water vapour observations. Since the measurements are taken as water vapour mixing ratio and total water mixing ratio per
volume they are converted to specific humidity $Q_v$ and cloud water content $Q_c$, respectively, for comparison with model data.

[1]Information obtained at https://www.dlr.de/fb/en/desktopdefault.aspx/tabid-3718/5796_read-8410/, visited on 19 June 2020





## 2.2 HyMeX Raman lidar

Data collected during the HyMeX SOP I by the Raman lidar system BASIL from the University of Basilicata (Ducrocq et al., 2014) is used (Di Girolamo et al., 2009). The instrument was positioned at the HyMeX 'supersite' in Candillargues near Montpellier (4.07°E, 43.61°N) on the French Mediterranean coast. BASIL provides profiles of water vapour starting from 60 m above the ground up to the lower stratosphere in steps of 30 m. A vertical 300 m running averaging is applied to each profile as well as a 1-hourly time running mean to account for comparison with the coarser resolved model data. Considering an integration time of 1 hour and a vertical resolution of 300 m, the statistical uncertainty affecting daytime water vapour mixing ratio measurements is found to be smaller than $0.5 \, g \, kg^{-1}$ (or 10%) up to 2.5 km and smaller than $3 \, g \, kg^{-1}$ (or 40%) up to 4 km. Nighttime performance is characterised by much smaller uncertainties, with a random error affecting vapour mixing ratio measurements not exceeding $0.01 \, g \, kg^{-1}$ (or 0.5%) at 4 km and of $0.035 \, g \, kg^{-1}$ (or 7%) at 10 km. The procedure applied to calibrate water vapour mixing ratio profile measurements by BASIL was described in detail by Di Girolamo et al. (2016). During HyMeX SOP I, BASIL was calibrated based on the comparison with radiosondes launched from the lidar site, with the launching facility being approximately 100–150 m away from the lidar system (Di Girolamo et al., 2016). More specifically, the mean calibration coefficients were estimated by comparing BASIL with the radiosonde data at all times when both systems were simultaneously operated (for a total of approximately 50 comparisons). The uncertainty affecting the calibration constant is estimated to be 3%. Only data outgoing from the instrument upward until a relative error of 100% is reached are used, that is up to 5–6 km in daytime and up to the tropopause at night. For the purpose of this research effort, profiles up to an altitude of about 3000 m, or 700 hPa, are considered. BASIL measured continuously between 21:44 13 October until 19:03 14 October (Di Girolamo et al., 2016).

## 2.3 Monte Lema radar

The Monte Lema radar is part of the Swiss network of C-band (5.5 GHz) Doppler weather radars operated by MeteoSwiss and it is located on the southern slopes of the Alps at 8.83°E, 46.04°N, and 1626 m altitude. Monte Lema was upgraded in 2011 to dual polarisation which, together with other technical enhancements, provided increased data quality (Germann et al., 2015; Gabella et al., 2017). The volume scan geometry comprises 20 elevations from -0.2°to 40°with a maximum range of about 18 km vertically and 246 km horizontally, while the interleaved scan strategy provides every 5 min a new full volume scan. The methods for reading and processing of the raw polar data from Monte Lema radar can be read in Figueras i Ventura et al. (2020).

## 2.4 Tracer experiment

A tracer experiment was performed using the perfluorocarbon tracer system PERTRAS. This instrument was newly designed for studying Lagrangian long-range transport and related dispersion of chemical species, e.g. in the Asian Monsoon system (Ren et al., 2015). Perfluoromethylcyclopentane (PMCP C6F12) has very low background concentrations in the atmosphere of 6-7 ppqv in Europe (Watson et al., 2007) and is thus very well suited for a tracer experiment. For the T-NAWDEX-Falcon





campaign, the release unit of PERTRAS for the inert tracer gas was installed on board of the light aircraft Partenavia P68 operated by enviscope[2]. The sampling unit of PERTRAS, on the other hand, was installed on board the Falcon. The intention

to separate the release unit from the sampling unit was to exclude possible contamination throughout the re-sampling of the tracer and to meet the logistic demands given by the relatively fast transport of air masses between the WCB inflow and outflow stage. The Falcon installation allowed to collect probes of air at an interval of about 5 min yielding a spatial resolution of 50 km. Such an experiment could be conducted only once during the T-NAWDEX-Falcon campaign and had to be carefully planned based on the operational ECMWF forecasts (Schäfler et al., 2014). During IOP 2 on 14 October 2012 suitable conditions

were given, the Partenavia was therefore transferred to Southern France and released in the morning hours between 09:09 and 09:39 UTC 6.2 l of the tracer in the target region. The liquid PMCP tracer was dispersed using a spray nozzle connected to the tracer reservoir and mounted outside the window of the Partenavia. Thereby, PMCP droplets with diameters less than 20 mm were released which evaporated readily. A day later, on 15 October, two Falcon flights (24 and 30 h after the release) above Germany were devoted to resample the tracer in the ascending air masses. For the tracer sampling, 64 adsorption tubes were

used where the PMCP molecules were trapped and concentrated. The samples were analysed after the flights in the laboratory using a gas-chromatic method described in Ren et al. (2014). Unfortunately, a technical issue concerning the manual time adjustment of the device likely occurred during the first sampling flight IOP2b.

## 2.5  Model data

For the calculation of WCB trajectories, we use the Ensemble of Data Assimilations (EDA) dataset from the European Centre

for Medium-Range Weather Forecasts (ECMWF). The EDA represents the best possible estimate of the state of the atmosphere considering uncertainties associated with observations and the ECMWF data assimilation system. It became operational in 2010, and in 2012, it became available as a set of 11 analyses that primarily serves to improve the initial conditions of the ensemble prediction system (Isaksen et al., 2010), with one control analysis (CTL) and 10 perturbed members. The slightly differing analyses are obtained by perturbing atmospheric observations, sea surface temperature fields and model physics in

the 4D-Var data assimilation cycle. The original resolution of the EDA of T399L91 in 2012 is interpolated to a 0.5°horizontal grid on the original 91 model levels. In addition to the 6-hourly standard EDA times, the intermediate products at 03, 09, 15, and 21 UTC are also used to increase the time resolution.

## 2.6  Trajectories and WCBs probabilities

Kinematic offline trajectories are calculated using the Lagrangian analysis tool LAGRANTO (Wernli and Davies, 1997;

Sprenger and Wernli, 2015). Here, we use trajectories to detect WCBs as well as matches between measurements in each member of the EDA, and they are therefore a central part of this study.

To know where WCB air masses ascend during the IOP, trajectories are started from the grid points between 1025 and 700 hPa with vertical steps of 25 hPa between 1000 and 100 hPa. They are initialised every 3 h over several days and calculated 48 h forward in time with trajectory positions saved every 1 h. To identify WCB trajectories, only the ones are selected that

---

[2]http://www.enviscope.de/enviscope_new/operation/airborne-platforms/partenavia-p68





ascend at least 600 hPa in 48 h (Madonna et al., 2014b). WCB probabilities are calculated using the WCB trajectories calculated
for all 11 EDA members based on the method that has been adopted from Schäfler et al. (2014) and Rautenhaus et al. (2015).
More specifically, the trajectories are disassembled into individual air parcel positions and then, for a given time, sorted into grid
boxes with a horizontal dimension of 0.5° in the horizontal and 25 hPa in the vertical. For each grid box, the WCB probability
(in %) then corresponds to the relative number of EDA members that have at least one WCB trajectory in the box. The method

results in a time sequence of Eulerian three-dimensional fields of WCB probabilities.

EDA-based trajectories are also used to identify Lagrangian matches between aircraft measurements at one time and any
type of measurements at an earlier or later time. To this end, forward and backward trajectories are calculated in each member
of the EDA, starting every minute from the respective flight route (coloured example trajectories in Fig. 1). The time range of
the trajectories is backward to 12 UTC 13 October and forward to 12 UTC 16 October to cover all measurements and the entire

WCB ascent. In this set of trajectories, those that fulfil the ascent criterion described above are labelled as WCB trajectories.
Similar to the method to obtain WCB probabilities, the positions of the EDA trajectories at a time of interest are assigned to
grid boxes (grey boxes in Fig. 1). Trajectory positions are hence available as a time sequence of Eulerian three-dimensional
probability fields (number of contributing EDA members; boxes with different grey shadings in Fig. 1). Matches between
measurements along the aircraft track from which the trajectories were started and any other measurements occur when the

trajectory probability (calculated with trajectories that started from the flight track) in the grid box that corresponds to the time
and location of the other measurement is larger than zero (the observing instrument in Fig. 1, which, in the example shown,
diagnoses differing matching probabilities with altitude). Hence, this procedure to identify matches accounts for the exact time
(of the measurement) but with a tolerance in space, represented by the 0.5° grid box around the exact trajectory position. If
the trajectory probability is high, then the Lagrangian match between the measurements can be regarded as more 'certain'.

If a matching trajectory also fulfills the WCB criterion, then we refer to this as a WCB match, i.e., then we know that two
measurements at different times and locations were sampling the same air parcel and that this air parcel was actually part of a
WCB. These WCB matches are of particular interest for our study.

## 3    Synoptic situation

The two-day IOP2 of the T-NAWDEX-Falcon campaign was planned to take measurements along a WCB ascending from the

western Mediterranean over the Alps towards the Baltic Sea. The WCB was induced by a low pressure system that originated in
the central North Atlantic before 13 October 2012. WCB air masses start from the warm sector already during this earlier phase
of the cyclone while it moves eastward, steered at the southern flank of a complex upper-level PV cutoff (Fig. 2a, b, the cyclone
is marked by a red 'L' in a). Downstream of the cyclone a zonal wind direction prevails at low levels over France at 18 UTC
13 October (Fig. 2a). North of the Pyrenees and further east over the Mediterranean, the low-level flow is orographically

280    influenced and arrives with increased wind speed and reduced humidity at the northwestern Mediterranean, as reflected by low
equivalent potential temperature ($\Theta_e$) values. This local wind, called 'Tramontane', is subject of a case study by Di Girolamo
et al. (2016) as part of the HyMeX campaign. They use observations of the HyMeX lidar that started operating at 21:44 UTC





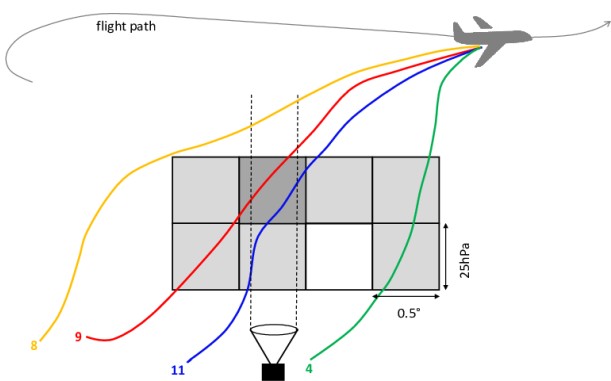

**Figure 1.** Schematic of the matching method: Backward trajectories from the aircraft calculated in different EDA members (coloured trajectories), grid boxes with contributions from different numbers of EDA members (grey boxes; corresponding to match probabilities of 0–18% in this example, 18% because 2 out of 11 ensemble members indicate a match), and a remote sensing instrument observing the match probability at that certain time step (black icon at the bottom).

13 October (see westernmost red marker for the lidar in Fig. 2). Within the westerly flow, WCB trajectories originating over the Atlantic are embedded that start to ascend prior to passing the lidar (Fig. 2b). At the same time, another WCB branch north
of the one from the Atlantic starts to rise over France. These two northern WCB branches with inflow from the Atlantic are directly steered by the surface cyclone that enters France from the Bay of Biscay at 06 UTC 14 October (not shown). The cloud band related to this ascending part of the WCB is visible east and north of the cyclone centre few hours later in the satellite image (Fig. 2d and black marker at 12 UTC 14 October in Fig. 2b).

    The cold front of the cyclone is located over the northwestern part of the Iberian peninsula and western France at 12 UTC
14 October, when the wind direction over the northwestern Mediterranean turned to a moist southwesterly flow onshore (Fig. 2c). In this flow, ahead of the surface cold front, another WCB branch is discernible at low levels (Fig. 2b) while this region is covered by heavy convective clouds in the satellite image (Fig. 2b,d, see also Duffourg et al., 2018 about the associated heavy convective precipitation event). The inflow of this Mediterranean WCB branch passes the HyMeX lidar just southeast (Fig. 2b). On a smaller scale, the strong westerly wind ahead of the cold front initiates a secondary lee cyclone east of the Maritime
Alps in the Italian Piemont region (red 'L$_2$' for the mature lee cyclone in Fig. 3b).

    The preceding upper-level PV cutoff forms a streamer-like structure during the development of the main cyclone 'L' and subsequently extends southward over the western Mediterranean (Fig. 3b). The low-level lee cyclone 'L$_2$' intensifies due to the interaction with the upper-level PV streamer. The Mediterranean WCB branch shown in Fig. 2b moves northeastward ahead of the PV streamer. Between 00 and 06 UTC 15 October, the WCB branch passes cyclone 'L$_2$' (Fig. 3b) and impinges upon
the Alps where it is forced to ascend. There the WCB air stream crosses the Monte Lema radar in the early phase of the ascent (red marker in Fig. 3b). Within the next 6 hours, the Mediterranean WCB air parcels reach the middle troposphere and proceed with a moderate ascent rate over Germany towards the Baltic Sea where it approaches the Atlantic WCB branch. Along with the WCB, the cold front of the main cyclone 'L' proceeds into Central Europe. Together they form the stratiform and scattered



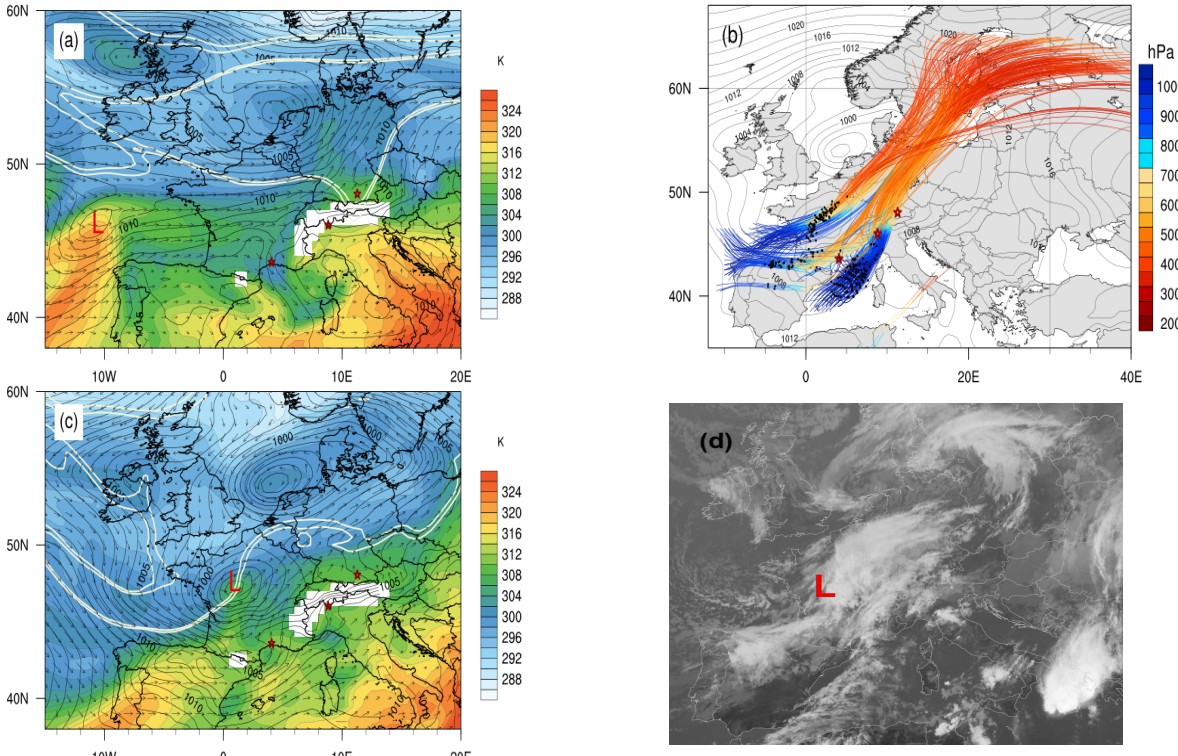

**Figure 2.** Synoptic development during the T-NAWDEX-Falcon IOP2: (a) EDA mean equivalent potential temperature at 850 hPa (colours, K), SLP (black contours every 1 hPa), wind arrows at 850 hPa (black) and PV at 315 K (white line of 1.5 and 2 pvu) at 18 UTC 13 October 2012. Red asterisks mark the measurement stations of the HyMeX lidar, Monte Lema radar and the air base of Falcon. (b) WCB trajectories of the EDA's CTL starting at 00 UTC 14 October 2012 (coloured by pressure, hPa) and SLP (black lines every 2hPa valid at 12 UTC 14 October), trajectory markers (black dots) for 12 UTC 14 October. (c) shows the same as (a) but for 12 UTC 14 October and (d) Meteosat infrared image at 12 UTC 14 October. The red 'L' in (a), (c) and (d) marks the location of the surface cyclone.

cloud band that is visible in the satellite image at 06 UTC 15 October (Fig. 3a). The research flight IOP2b was designed to climb
stepwise along the WCB ascent towards northeast Germany and then to cross the outflow in the area of northern Germany,
Poland and the Baltic Sea approximately in east-west direction (red/violet flight route in Fig. 3b). In the northernmost part, the
flight route traverses the centre of the surface cyclone. As with flight IOP2b, also flight IOP2c (blue flight route in Fig. 3b)
occurs in the region of the WCB and the cold front, which is crossed several times.

South of the Alps the upper-level PV streamer cuts off and the former lee cyclone 'L$_2$' evolves into a wide cyclone below the
PV cutoff (Fig. 3c). The initial cyclone 'L', however, weakens over the Baltic Sea and re-intensifies later below a newly formed
short-wave trough in the hours after the research flights. In the upper levels, the WCB outflow is directed northeastward by the
short-wave trough (Fig. 2b, Fig. 3c).





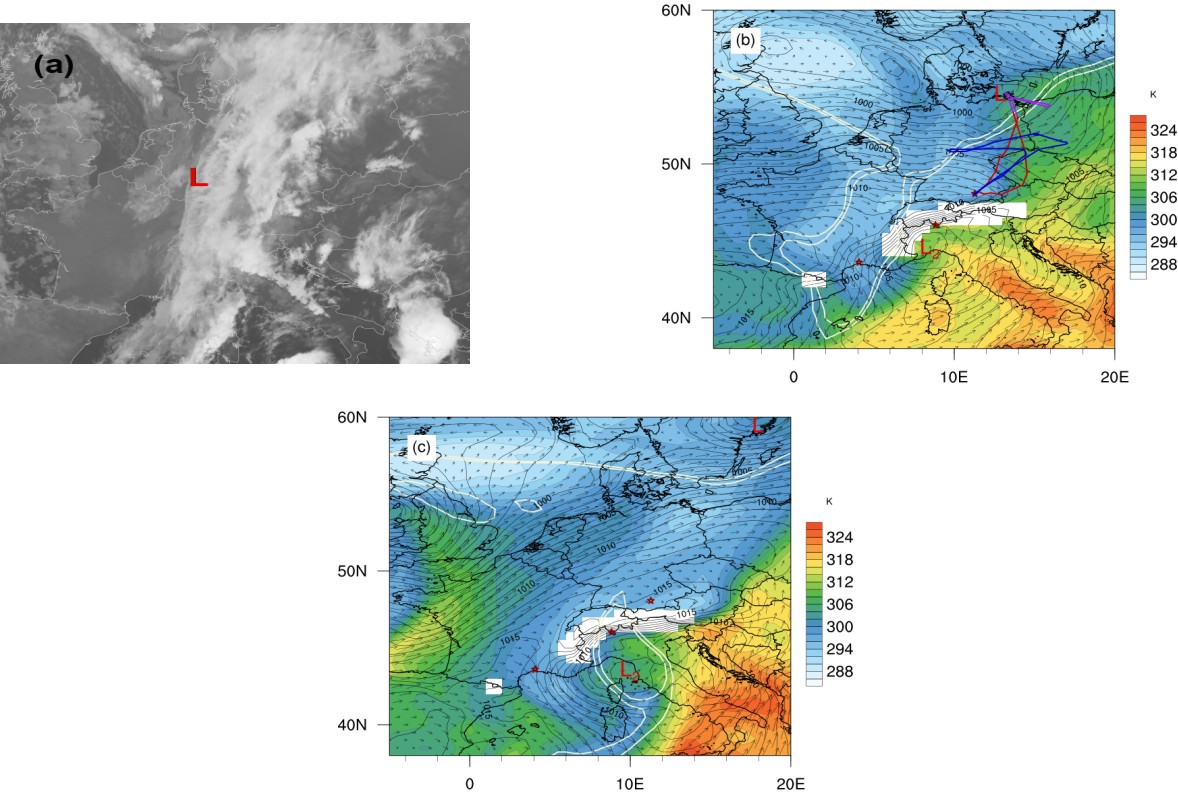

**Figure 3.** Continuation of the synoptic situation: (a) Meteosat IR image at 06 UTC 15 October 2012, (b) same as in Fig. 2a, but for 09 UTC 15 October and (c) 00 UTC 16 October 2012. In (b) flight IOP2b is shown in red with the violet part from 08:30 to 09:30 UTC, flight IOP2c in blue and the dark red 'L$_2$' marks the position of the secondary cyclone.

## 4 Results

### 4.1 Aircraft measurements

The two Falcon flights of the T-NAWDEX-Falcon IOP2 north of the Alps on 15 October 2012 are presented in the following two subsections. They form the starting points for investigating Lagrangian matches with further observations of this WCB in a later section. Here we elaborate where WCB air was transsected by the flights, and we compare the airborne in situ observations of water vapour and cloud condensate with the EDA.

#### 4.1.1 Flight IOP2b

The morning flight IOP2b on 15 October intended to follow the gradual mid- to upper-level ascent of the WCB from southern Germany towards the Baltic Sea. The aircraft took off in overcast conditions and ascended immediately towards mid-level WCB air between 700 and 500 hPa. Before entering this layer where WCB probabilities reached values up to 100% (Fig. 4a),





snow (SWC) peaks with negligible liquid water content (LWC) and cloud cover in the EDA at 07:36 UTC (Fig. 4b, red curve

in e), which indicates snow falling from the WCB into a subsaturated region (not shown) where it likely sublimates. Exactly

there, water vapour in the EDA is distinctly lower than the measured value, as shown by a downward deflection of the relative

error of the EDA's moisture with respect to the measured values (Fig. 4d, red curve).

Subsequently, where the aircraft intersected the WCB with the maximum probability at 07:42 UTC at an altitude of 560 hPa,

the EDA consistently shows a peak of 100% cloud cover (Fig. 4a,c and red line in e). The EDA cloud condensate along the

WCB consists of high values of SWC, LWC, and also ice (IWC) water content (Fig. 4b). LWC and IWC peak at the time of

100% cloud cover. The in situ measurements of water vapour and cloud condensate $Q_c$ confirm the moist and cloudy conditions

in this early flight period, with mean observations in the range of the interpolated values from the EDA (compare black and

grey lines in Fig. 4d,e). A distinctly greater peak of $Q_c$ is visible in the native resolution of measured $Q_c$ that coincided with

the highest WCB probability and the peak in cloud cover. Note that for fair comparison, cloud condensate $Q_c$ in the EDA is

here calculated from the non-sedimenting cloud species LWC and IWC alone (i.e. without SWC) since the Waran instrument

most likely did not efficiently collect sedimenting snow particles. According to the slope of the 0°C isotherm (black line in

4a), the aircraft ascends from behind the surface cold front into the warm sector of the cyclone in the mid-troposphere, which

is a typical region for WCB ascent.

After the initial climb, the aircraft continued with a 35-min quasi-horizontal leg at 500 hPa (from about 07:45 to 08:30 UTC).

According to most of the EDAs, the flight is first located along the upper edge of the WCB and then outside (Fig. 4a). For these

air masses, the maximum 48 h ascent varies between more than 600 to 300 hPa (Fig. 4c). Specific humidity agrees fairly well

between EDA and observations in this period (Fig. 4d). The concentration of the cloud species in the EDA are much lower than

before when crossing the WCB (Fig. 4b). The measurements reveal that $Q_c$ goes down to (near) zero during this period, while

the EDA values show small but elevated condensate of IWC and some SWC (Fig. 4b,e).

The subsequent step-wise ascent of the aircraft leads to intersections with the WCB outflow between 450 and 350 hPa, where

WCB probabilities reach up to 70% at 08:40 and from 09:00 to 09:45 UTC (Fig. 4a). There, the aircraft is on its northernmost

leg and crosses the WCB first eastward and later westward (Fig. 3a) with the turning point at 09:15 UTC. Near that time,

increased WCB probabilities extend vertically from 850 to 350 hPa. At the level of the aircraft the maximum ascent of the

observed air mass is highly uncertain and varies between 300 and 700 hPa in 48 h (Fig. 4c). Cloud condensate values in EDA

are rather low (Fig. 4b) and smaller than those measured. This is similar for water vapour where EDA values are also lower

than the ones observed (Fig. 4d,e).

After a high-altitude southward transfer, the aircraft started to descend at about 10:25 UTC to 700 hPa while crossing again

to the cold side of the surface front at 10:30 UTC. Thereafter, WCB probabilities reach up to 30% and EDA show high values

of LWC and SWC, indicating supercooled liquid water in the mixed-phase cloud at the lower edge of the WCB (Fig. 4a,b).

This is very similar to the intersection of the lower edge of the WCB in the beginning of the flight at around the same pressure

level. Again, the EDA values of cloud condensate are larger than in any other flight period, but here they are slightly higher

than the observations (Fig. 4e). For water vapour the agreement is better (Fig. 4d), but again shows reduced values in the EDA

where high SWC coincides with low LWC and cloud cover below the WCB between 10:43 and 10:47 UTC.





**Figure 4.** Time series of measurements and interpolated EDA fields along the Falcon flight IOP2b from 07:34 to 10:52 UTC on 15 October 2012. (a) WCB probabilities (colours, in %) in the vertical column along the flight track, pressure of the aircraft altitude (dark red line, in hPa), and EDA-mean temperature (0 and $-38°$C, black lines); (b) EDA cloud species (see legend, in $\mathrm{mg\,kg^{-1}}$); (c) maximum pressure decrease (ascent) within 48 h of the air parcels that were measured during the flight, calculated for each EDA member (left y-axis) and, as in (a), flight altitude (in hPa, right y-axis); (d) specific humidity $Q_v$ (in $\mathrm{g\,kg^{-1}}$) from in situ measurements (black line for 5 min running mean and native resolution as dots) and from EDA (grey line for mean and shading between minimum and maximum EDA values), relative error of the EDA members compared to the measurements (red line and shading, right axis); and (e) cloud condensate $Q_c$ (in $\mathrm{g\,kg^{-1}}$, from in situ measurements (black line and dots as in (d)) and EDA consisting of liquid and ice water content (grey), and EDA cloud cover (red, right axis).



In summary, Falcon flight IOP2b intersected the WCB with a high probability in the mid-troposphere during the ascent and at its outflow level. However, two horizontal legs (before 08:00 and after 10:30 UTC) just missed the center of the WCB as they

were slightly too high and too low, respectively. In all clouds within or near the WCB the observed cloud condensate agrees well with the EDA values. According to the EDA, the lower part of the WCB contains supercooled liquid water at temperatures between −5 and −10°C. As found by radar observations in Gehring et al. (2020), the formation of supercooled liquid water in the phase of strongest WCB ascent facilitates aggregation and riming, and provides ideal conditions for rapid precipitation growth.

### 365 4.1.2 Flight IOP2c

The flight in the afternoon of 15 October again went towards the Baltic Sea (see blue line in Fig. 3a) and crossed the cold front with WCB probabilities up to 40% at around 550 hPa (Fig. 5a). As for the previous flight, the ascent across the WCB leads to peaks of first SWC (at the lower edge of the WCB) and then LWC and IWC (within the WCB) according to the EDA (Fig. 5b). Here, water vapour is underestimated in EDA compared with the measurements and shows a local minimum where snow

is likely to sublimate below the WCB as already seen in the flight before (Fig. 5d,b).

The subsequent flight leg at 350 hPa is located just above an extended region with high WCB probabilities. The observed air ascended by 300 hPa within 48 h according to Fig. 5c. The measured specific humidity in EDA is again too low (Fig. 5d). Given the moderate ascent of this air and its location just above the WCB, it is plausible that this cloud layer corresponds to a so-called in situ cirrus, which often tops the liquid-origin cirrus produced by the strongly ascending WCB (Wernli et al., 2016).

Shortly before 15:00 UTC, the aircraft descended to 500 hPa and, according to the EDA, intersected a region with WCB probabilities up to 100% (Fig. 5a). Along this flight leg between 15:00 and 15:25 UTC, SWC and IWC are elevated in EDA (Fig. 5b). The observed water vapour is still higher than in the EDA, but with a smaller deviation than before. The bump of the 0°C isotherm below the aircraft in Fig. 5a indicates that the aircraft crossed the surface cold front and entered the warm sector at 15:20 UTC before crossing the cold front again in the reverse direction at 15:35 UTC.

Taking both flights (IOP2b and 2c) together, the aircraft sampled WCB air with high probability during the ascent and outflow of the WCB on several legs of the flights. Whenever high WCB probability was intersected, specific humidity and cloud condensate in EDA are increased. The magnitude and structure of cloud condensate is well represented in the EDA compared with the observations. Water vapour is often underestimated, in particular below increased WCB probability where precipitation is likely to sublimate or evaporate. Short periods with overestimated water vapour in EDA occurred near regions with large

gradients of WCB probabilities, i.e. where the aircraft encountered regions with lower probability of WCB occurrence in the EDA and where, in reality, most likely large gradients in humidity occurred.

### 4.2 Lagrangian matches of the aircraft-probed WCB air masses with ground-based measurements

In this section, the pathway of the aircraft-observed WCB air, i.e., of WCB trajectories that intersected the flight route, is considered in more detail. In addition, where possible, ground-based water vapour lidar and precipitation radar measurements

south of the Alps are considered, which, unplanned, sampled some of the WCB air masses during the inflow and ascent phase.







**Figure 5.** Same as Fig. 4 without panel e, but for the Falcon flight IOP2c.



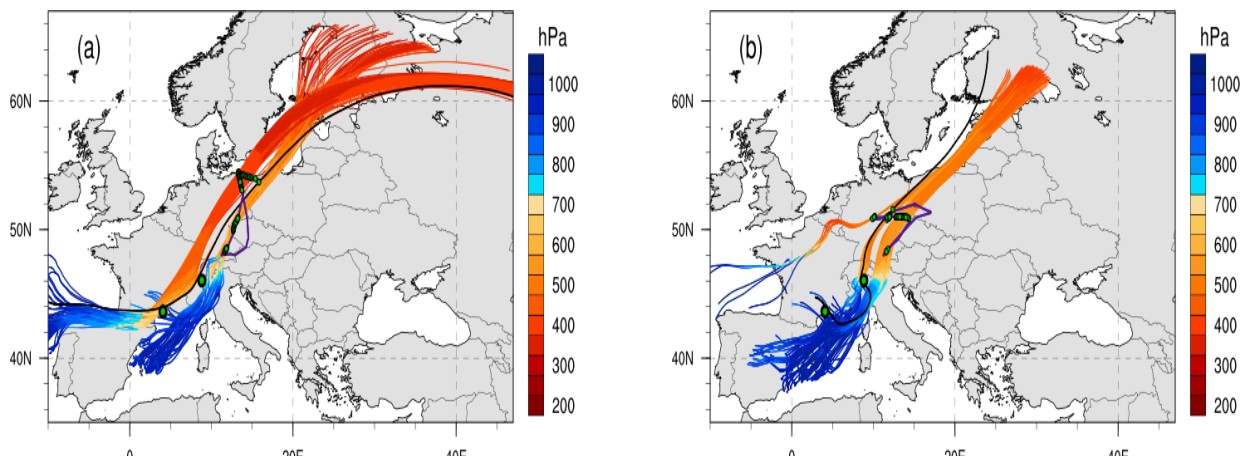

**Figure 6.** (a) Forward and backward trajectories from all EDA members started from the IOP2b flight route (green line), which fulfil the WCB criterion (coloured with pressure, in hPa); locations where the WCB trajectories were observed are shown by green markers (see text for details); (b) same for IOP2c. Trajectories are shown for the time interval 12 UTC 13 October to 06 UTC 16 October. In each panel, the black trajectory marks the one discussed in detail in Fig. 7.

Both Falcon flights presented in the previous section encountered different WCB branches, which originated either from the North Atlantic or the Mediterranean (Fig. 6). The two WCB intersections during IOP2b in the mid and upper troposphere, respectively, are related to these two branches, as shown in Fig. 6a, for the WCB trajectories that were sampled by the Falcon flight. The branch with inflow from the North Atlantic starts ascending towards the east over the Pyrenees, passes the HyMeX

lidar station at the Mediterranean coast (southernmost green dot in Fig. 6a), enters the domain of the Monte Lema radar at the southern slope of the Alps (green dot further northeast in Fig. 6a), and is eventually observed by the Falcon during the northernmost leg of flight IOP2b from 08:30 to 09:30 UTC at 350 hPa (northernmost green dots in Fig. 6a, Fig. 4a). The Mediterranean WCB branch, however, crosses the HyMeX lidar at very low levels during its inflow period (at least some of its trajectories) and then starts ascending south of the Alps where it is observed by the Monte Lema radar. After crossing the

Alps, this WCB branch is then sampled by the Falcon in the mid-troposphere at 650-500 hPa (see green dots in Fig. 6a over southern Germany). The ascent of this branch occurred more steadily and the outflow reached further north, compared to the North Atlantic branch.

The WCB observations during IOP2c are dominated by the Mediterranean branch (Fig. 6b) with an ascent comparable to the one described before. Also this branch was (partially) observed by both ground-based instruments before being sampled by

the Falcon between 600 and 500 hPa (Fig. 6b, Fig. 5a). During this flight, only few WCB trajectories with inflow from the Bay of Biscay were observed by the Falcon. However, a small bundle of WCB trajectories from this region ascended much further north compared to the rest of the WCB, presumably mainly lifted by the cold front and without an orographic influence. These trajectories were measured by the aircraft at the westernmost tip of the flight on 500 hPa at 14:50 UTC.



From the WCB trajectories that "matched" with the lidar and radar, i.e., intersected the vertical column above both ground-based instruments before being sampled by the aircraft, two are selected as exemplary WCB trajectories and referred to as trajectory T1 and T2, respectively. T1 represents the WCB branch with inflow from the North Atlantic that was sampled during flight IOP2b, and T2 the WCB branch with inflow from the Mediterranean sampled during flight IOP2c (black trajectories in Figs. 6a,b, respectively). We emphasise that these triple Lagrangian matches along the WCB trajectories between lidar water vapour, radar reflectivity and aircraft in situ measurements provide a rare opportunity to compare the evolution of humidity and clouds along a WCB in analysis data with independent observations. In the following two subsections, the ascent behaviour and cloud evolution along the selected North Atlantic and Mediterranean trajectories T1 and T2 are discussed.

### 4.2.1 WCB ascent from the North Atlantic

Trajectory T1 with inflow over the North Atlantic started to ascend gradually in the late hours of 13 October to the west of continental Europe (Fig. 7a). The initial ascent goes along with the formation of liquid cloud water and later rain as the lifting continues north of the Pyrenees at about 06:00 UTC 14 October (Fig. 7a). After 09 UTC 14 October, when the trajectory rises from 800 to 700 hPa, snow falls from the vertically deep cloud to the height of the trajectory, which experiences still temperatures above the freezing level. Just after 12:00 UTC, the trajectory crosses the 0°C isotherm and melting of snow leads to intense rain beneath the WCB, which at that time crosses the western part of the French Cevennes mountains. The trajectory T1 is at 630 hPa when it passes the location of the lidar near Montpellier in Southern France at 16:00 UTC (grey bar in the middle panel of Fig. 7a). Until that time, the trajectory has lost about 75% of its initial water vapour of 12 g kg$^{-1}$ due to cloud condensation. Because of the clouds associated with the ascending WCB trajectory, the lidar could only provide reliable water vapour profiles below the level of the WCB. In this layer from the surface to about 700 hPa, characterised by southerly flow, the lidar reveals a relatively uniform moist layer with specific humidities of 5-6 g kg$^{-1}$ (Fig. 7a, between 16:00-19:00 UTC). The comparison with EDA shows that the analyses are too moist in the lower part of this layer and too dry in the upper part, each by about 1 g kg$^{-1}$. In fact, during this period with the overrunning WCB (shown by black contours for WCB probability and yellow hatching for matches with IOP2b in Figs. 6a, 7a), most likely the humidity profile in this layer is strongly influenced by below-cloud evaporation of rain. For instance, if the model is assumed to underestimate precipitation from the WCB, this might explain the negative humidity bias in the 150 hPa thick layer just beneath the WCB.

In the evening of 14 Oct, T1 further ascends over the western Alpine range and after a short break of the continuous ascent, it rises even more steeply above Southern Switzerland where it is observed by the Mont Lema radar at about 00:00 UTC 15 October (green dot in Fig. 6a). According to the EDA, the WCB at the position of T1 at 470 hPa still contains some supercooled liquid water, ice and snow (lower panel in Fig. 7a). In the radar measurements, the intersection point of the trajectory is shown by the red asterisk at about 8.4°E in Fig. 8b at the upper edge of the layer with increased WCB probabilities (the layer with maximum WCB probability is composed of WCB trajectories with inflow from the Mediterranean and the Atlantic). Near the location of T1, i.e., where the cloud consists mainly of snow and ice according to EDA, radar reflectivities are between 0-10 dBZ. According to the radar, the precipitating cloud extends up to about 400 hPa. This uppermost part of the precipitating cloud is likely produced by air that is lifted on top of the WCB and does not meet the WCB ascent criterion. Below the position



of T1, reflectivities increase markedly in the layer with high WCB probabilities between 900-600 hPa. There is an indication of a bright band due to snow melting near 750 hPa with moderate to high reflectivity values exceeding 30 dBZ. This points

to intense rainfall associated with the WCB crossing the Alps. In contrast, most of the non-WCB air that was later measured by Falcon (blue asterisks in Fig. 8b) appears to be dryer and intersects the radar scan where radar reflectivity shows a local minimum.

As shown by the vertical cross section along the trajectory in Fig. 7a, the cloud in EDA is particularly deep when it passes the high mountains near the radar, extending from the surface to about 300 hPa. Thereafter, the trajectory continues to ascend gently

and comparatively thin clouds are still present when it is measured by the aircraft at 08:54 UTC 15 October on the 400 hPa level near the northeasternmost point of the flight (Figs.6a, 7a, 4). The trajectory then slowly approaches the tropopause, about 48 h after the start of the ascent (Fig. 7a).

### 4.2.2 WCB ascent from the Mediterranean across the Alps

The air parcels of trajectory T2, which is later measured during flight IOP2c, initially moves over the Massif Central towards the

Mediterranean (Fig. 6b). Just before reaching the coast, the low-level trajectory is sampled by the lidar near Montpellier after 21 UTC 13 October (grey bar in the middle panel of Fig. 7b), which corresponds to the beginning of the lidar measurements on this day (Fig. 8a). According to the lidar, T2 arrives with relatively low specific humidity values of about 4 g kg$^{-1}$, which is distinctly dryer than the southerly flow some hours later. The EDA has a relatively large moist bias in the near-surface layer close to the WCB trajectory T2 (see yellow circles in Fig. 8a) of about 2 g kg$^{-1}$. This low-level surplus of moisture in EDA

persists for most of the 24-h measurement period with only a short interruption during another episode with elevated WCB probabilities at 12 UTC 14 October. The moist bias in EDA does not seem to depend on the large-scale flow direction.

After crossing the coast, the humidity along T2 increases (Fig. 7b) while the trajectory makes a long turn over the northwestern Mediterranean until 16 UTC 14 October (Fig. 6b). It moistens markedly due to ocean evaporation, reaching a maximum of 8 g kg$^{-1}$, which is, however, still much dryer than the initial humidity of T1 (Fig. 7a,b). According to the study by Rain-

aud et al. (2016) as part of HyMeX, the fueling of the WCB with sea moisture coincided with an evaporation event that was dominated by strong low-level wind and to a lesser degree by the low-level gradients of temperature and humidity.

At about 21 UTC 14 October, T2 reaches the southern slopes of the Alps where strong orographic lifting goes along with the formation of very dense clouds. Values of cloud condensate at the trajectory's position, in particular SWC, are distinctly larger than at any time during the ascent of T1.

Close to the time when peak values of SWC occur in EDA, i.e. shortly after 06 UTC 15 October, the trajectory passes through the vertical cross section of the Monte Lema radar reflectivity shown in Fig. 6b at 8.5° E and an altitude of 650 hPa (see red asterisk in Fig. 8c). T2 is located right above the well-defined melting layer. The radar reflectivity at the location of T2 is about 25 dBZ consistent with the high values of precipitating snow (>500 mg kg$^{-1}$ in EDA). The trajectory is passing through a region of high WCB probabilities, and it is also accompanied by other WCB and non-WCB air that will later be

measured by the Falcon (red and blue dots in Fig. 8c).



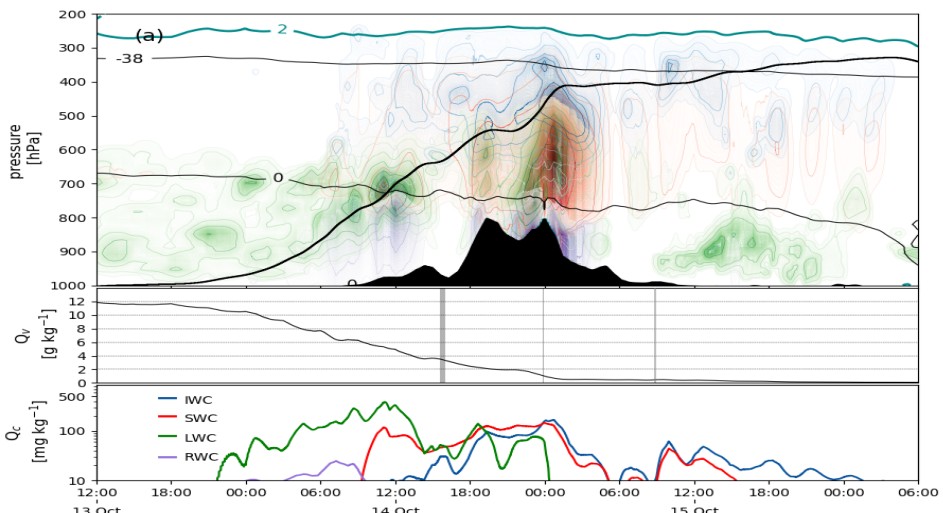

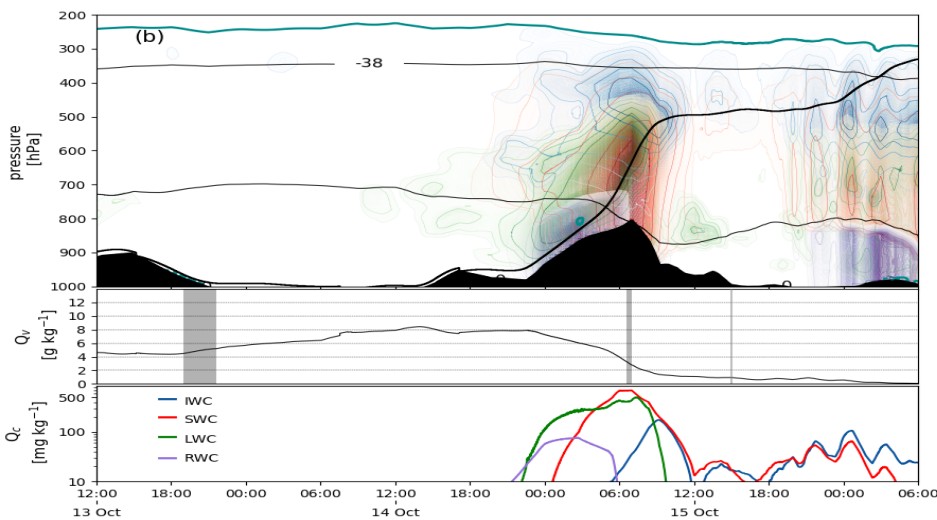

**Figure 7.** Vertical profiles along trajectories (a) T1 and (b) T2 (black lines in Fig. 6). Shown are in upper panels of (a,b) the height of the trajectory (thick black line), cloud species from EDA (RWC in purple with contours between 10-100 mg kg$^{-1}$, LWC in green with contours between 10-300 mg kg$^{-1}$, SWC in red with contours between 10-100 mg kg$^{-1}$, and IWC in blue with contours between 10-100 mg kg$^{-1}$); temperature (thin black lines for $0°$C and $-38°$C), and the 2-pvu tropopause (cyan line). The middle panels show specific humidity $Q_v$ along the trajectories and gray vertical bars indicate were measurements were taken. The lower panels show cloud species along the trajectories (colours see legend).


Compared to the time when trajectory T1 passed the radar about 7 hours earlier, the reflectivities are slightly higher and horizontally homogeneous during the crossing of T2. This might be related to the now more coherent flow of WCB air from the Mediterranean towards the Alps compared to the more complex flow situation during the passage of T1 (Fig. 6). For T2, cloud formation and rainout occur entirely at the Alps and no previous dynamical lifting reduced the load of water as in the case of T2 (Fig. 7).


The steep drop of the height of the 0°C isotherm in Fig. 7b at the time of maximum ascent of T2 indicates that the orographic lifting coincided most likely with enhanced dynamical forcing for ascent and was associated with the passage of the cold front. Again, supercooled liquid water exists in EDA up to about 500 hPa. After reaching this level, cloud condensate values decrease strongly and the ascent becomes very weak for more than 12 h. During this period, the Falcon flight intersects T2 at 15 UTC, when the aircraft enters the region of high WCB probabilities (Figs. 5). Later on 15 October, the air parcel of the trajectory is lifted further by another cold front near the Baltic Sea, discernible by deep clouds and the lowering of the 0°C isotherm in Fig. 7b. It reaches the tropopause about 30 h after the start of its ascent.


This detailed analysis of two prototype WCB trajectories reveals the complexity of this orographically influenced WCB, which originates from both the eastern North Atlantic and western Mediterranean and exhibits high variability in its ascent behaviour with a gradual slantwise, dynamically-driven ascent for T1 and a more step-wise and mixed orographically and dynamically-driven ascent for T2. For the latter, the very steep ascent when the cold front crosses the Alps almost reaches the threshold of 320 hPa in 3 h. Such rapid ascent has been used to identify embedded convection in WCBs by Oertel et al. (2020). It is during this period when the radar observes the largest near-surface values up to 40 dBZ, but here, the rather uniform structures in the radar reflectivity do not hint to embedded convection (Oertel et al., 2019).


### 4.3 Tracer experiment


An experiment with the physical release of a passive tracer by a small aircraft in the inflow of the WCB was conducted as part of the campaign. The main objective was to "label" WCB air in the inflow and then later "catch" the same air during its ascent by the Falcon aircraft with the objective to experimentally confirm the WCB ascent as determined by the model data. The use of a passive tracer enables, in principle, a validation whether the air observed at a later time by the Falcon actually had its origin in the labelled WCB inflow region. Here we report about the outcome of this challenging experiment, which essentially relies on exact forecasts and an effective flight planning.


The release flight of the tracer gas occurred between 09:00 and 10:00 UTC on 14 October off the Mediterranean coast near Marseille (light red marker in Fig. 9b near 6°E). The small aircraft Partenavia (see Section 2.4) flew through the near-surface air with WCB probabilities exceeding 70% at about 950 hPa (Fig. 9a). After a short transfer the release of the tracer started at 09:09 UTC at a level of 970 hPa (marked by the thick red line in Fig. 9a). Then the aircraft ascended continuously while spraying. It left the WCB inflow after about 15 min and continued to realease tracer for another 15 min in air that will ascend by more than 400 hPa within the next 48 h (lower panel in Fig. 9a). The release ended at 09:39 UTC on 750 hPa.


Trajectories are calculated from the tracer release section of the flight in all EDA members, and also the subset of these that fulfil the WCB criterion is selected. We applied the same method as for the WCB probabilities (section 2.6) to obtain

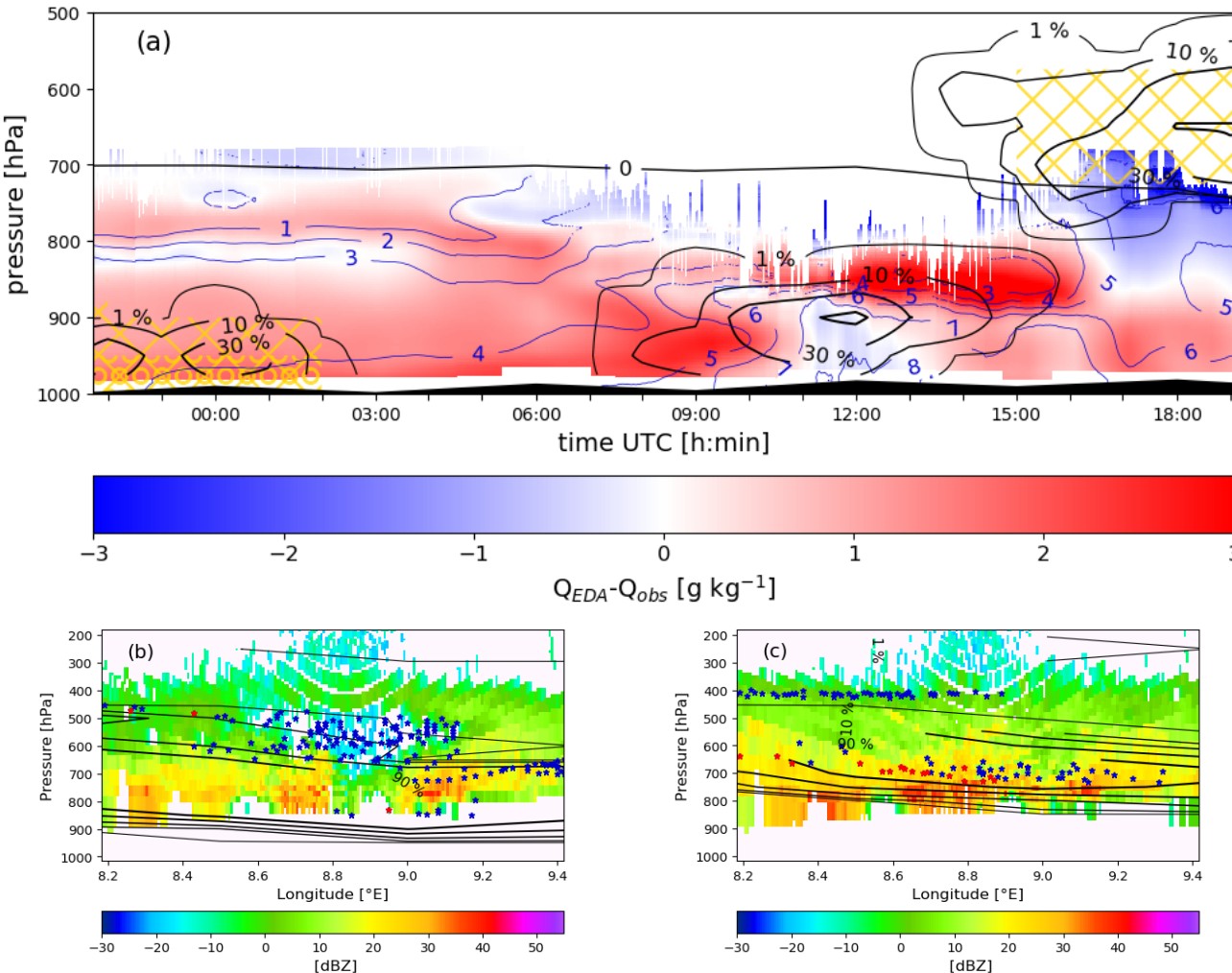

**Figure 8.** Time-height plots of measurements from (a) the water vapour lidar near Montpellier and (b) and (c) the Monte Lema radar. Panel (a) shows the difference of specific humidity between the EDA mean and the lidar observations (colours, in g kg$^{-1}$), lidar-observed specific humidity (blue contours, in g kg$^{-1}$), the EDA mean 0°C isotherm (thin black contour), WCB probability (thicker black contours for 1, 10, 30, 60 and 90%), and intersections of air parcels that were later sampled by the Falcon (yellow hatching, crosses for flight IOP2b and circles for IOP2c) from 21:44 UTC 13 October to 19:03 UTC 14 October 2012. Panels (b, c) present vertical cross-sections of radar reflectivity from the Monte Lema radar (colours, dBZ) at 46.15° which is 12 km north of the radar, WCB probabilities (black contours, as above), and intersection points of Falcon-probed trajectories (blue) and Falcon-probed WCB trajectories (red), at (b) 23:40 UTC 14 October with Lagrangian matches with flight IOP2b and (c) at 06:35 UTC 15 October 2012 with Lagrangian matches with flight IOP2c.




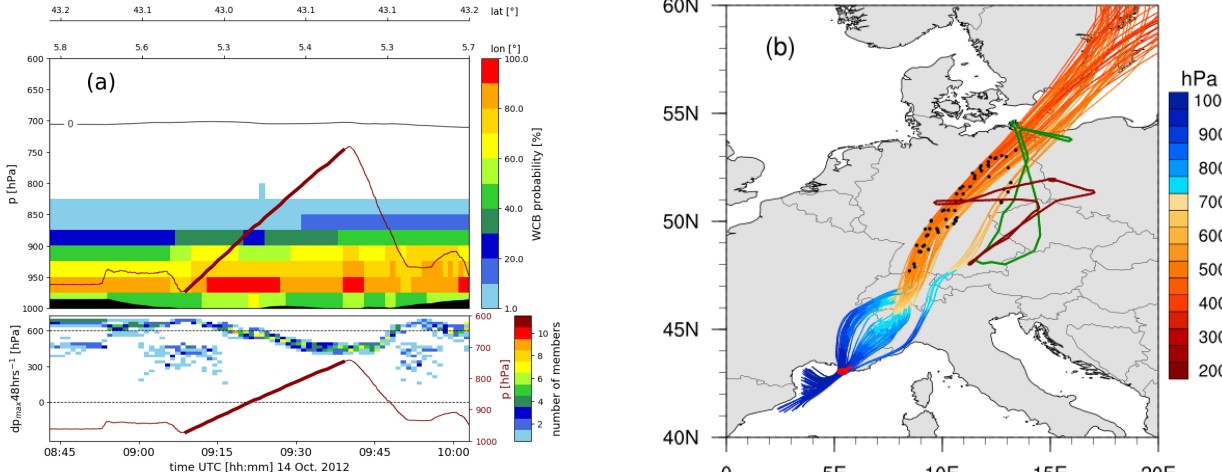

**Figure 9.** (a) Vertical cross section along the tracer release flight: WCB probability (colours), flight altitude (red line), tracer release (thick part of the red line) and temperature (black line, 0°C). The lower panel shows the same as Fig. 4c, here for the tracer release flight. (b) WCB trajectories started from the tracer release section (thick red line in a), trajectories coloured according to pressure, tracer release flight route (bright red), Falcon flight IOP2b (green), Falcon flight IOP2c (dark red) and position of the trajectories at 09 UTC 15 October (black dots).

probabilities for the occurrence of the tracer air mass, which we refer to as tracer probability. As shown in Fig. 9b, the WCB trajectories containing tracer gas move similarly across the Alps and have a comparable life cycle as the Mediterranean trajectory T2 discussed before (see Fig. 7b). In the evening of 15 October, WCB with tracer passes the radar below the Atlantic trajectory T1 at 700-600 hPa according to the model trajectories (not shown in Fig. 8b). In the following hours, the Falcon observations were conducted with one of the goals to take samples of the tracer gas.

The position of the WCB trajectories carrying the tracer at 09:00 UTC 15 October, i.e. at the time of flight IOP2b, is indicated by the black markers in Fig. 9b (flight route in green). Their positions hint to the proximity of the tracer and the flight route. Note that only WCB trajectories are displayed in the figure and that these are surrounded by more tracer-carrying air masses that ascend less strongly than the WCB. A vertical cross section along the Falcon flight reveals that the aircraft is at this time and during the previous hour at the upper edge of the tracer plume, indicated by relatively high tracer probability of up to 40%

in Fig. 10a. Hatching shows where tracer-carrying WCB trajectories intersect the flight curtain (however, with a low probability, not shown). The lower panel in this figure shows increased values of the tracer concentration measured in situ during the Falcon flight in the early part of the flight leg on 500 hPa. Comparison with Fig. 4a shows that the tracer-carrying air in Fig. 10a, which does mainly not fulfil the WCB criterion, fills the gap in the high WCB probability between 08:00 and 08:30 UTC there. Later the flight crosses low tracer probabilities until 09:30 UTC. The measurements corroborate the presence of tracer gas with

considerable high concentrations of up to 150 ppbv over a longer section along the flight. The measurements, however, do not exactly agree in time with the proposed peaks in tracer probabilities from the trajectories. The matching (and estimated tracer transport) based on kinematic trajectories comes with some uncertainties. For instance, mixing processes that, e.g., by deep convection in the WCB inflow (see Fig. 3a), might have led to significant dispersion, possibly leading to a slight shift of the



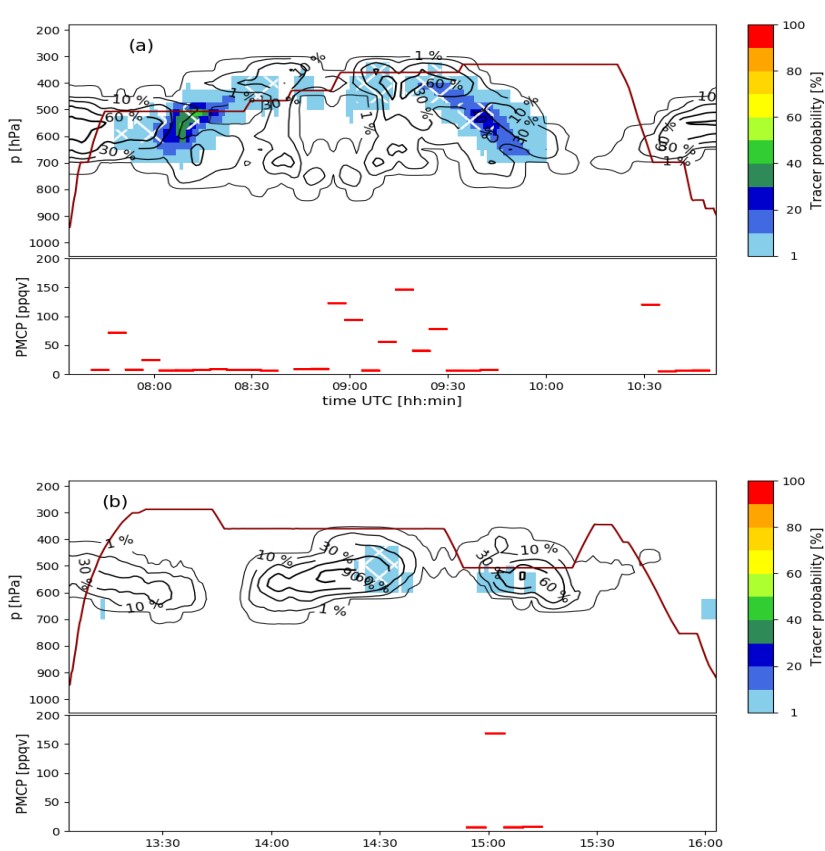

**Figure 10.** Tracer probability along the Falcon flights (a) IOP2b and (b) IOP2c. Upper panels show tracer probabilities (colours for all trajectories, white hatching for WCB trajectories), flight altitude (dark red line), WCB probabilities (black contours for 1, 10, 30, 60 and 90%), and the lower panels show the tracer concentrations sampled onboard the Falcon.

long-range transport in the atmosphere compared to the trajectories, which do not contain the effects of local-scale dispersion.

Also, a technical issue concerning the manual time adjustment of the device cannot be completely excluded. This issue might have been resolved for the subsequent Falcon flight IOP2c in the afternoon of the same day. High concentrations of observed tracer gas of 170 ppbv do exactly coincide with the only flight segment at 15 UTC (Fig. 10b) where tracer probabilities point to a long-range tracer transport to the flight track. The increased tracer probability for this flight agrees with the increased WCB probability in Fig. 5 and clearly points to a transport of the tracer gas with the WCB.

This confirms that we successfully sampled a tracer, which was released in the boundary layer over the Mediterranean Sea, 30 h later and 1000 km further north at an altitude of 500 hPa. This long-range transport occurred with the Mediterranean WCB air mass close to the evolution of the example trajectory T2 described in subsection 4.2.2, which realized triple Lagrangian matches with observations.





## 5  Summary and discussion

This detailed WCB case study revolves around a unique set of airborne in situ observations taken during the T-NAWDEX-Falcon field experiment that happened in parallel to the HyMeX SOP1 in autumn 2012. The key object of this study is an orographically influenced WCB ascending with two inflow branches across the Alps towards the Baltic Sea. The WCB shows a rather complex flow behaviour in the vicinity of the Alps. There is low-level inflow of air into the WCB from both the North Atlantic and the Mediterranean, where the former mainly flows around and the latter rises above the Alps, with some of the

lifting being clearly due to mechanically forced orographic ascent at the southern Alpine slopes. This is so far the most detailed WCB case study near complex topography, illustrating that dynamic and orographic lifting can interact in the formation and evolution of a WCB. Assessing the generality of this finding for other WCBs near mountains is left for further research. Two inflow branches with moisture sources in the North Atlantic and the Mediterranean, respectively, were also found in studies of heavy precipitation events on the Alpine south side (Winschall et al., 2012), which emphasises the potential of WCBs in

contributing to severe weather in this region (Buzzi et al., 1998). North of the Alps, where the two branches of the WCB further ascend towards the Baltic Sea, two research flights intersected both branches, one at mid-level during ascent and the other later in the outflow.

In order to clearly attribute certain periods of the in situ aircraft observations to the WCB, we used a sophisticated trajectory approach. Kinematic air parcel trajectories are subject to uncertainties in the wind field and to numerical errors in the trajectory

computation. To address these uncertainties and to provide more confidence about the identification of the WCB, we determined the WCB probabilistically by calculating trajectories in each of the 11 members of the ECMWF's ensemble data analysis (EDA) – an operational dataset that quantifies uncertainties in the representation of the state of the atmosphere.

Observational evidence for the long-range transport in the WCB was obtained from an airborne tracer release experiment. The experiment involved the release of an inert tracer gas in the WCB inflow region in the western Mediterranean by a

small aircraft, and the sampling of the tracer by the Falcon aircraft 30 hours later and 1000 km further north over Germany. The setup of this experiment was unique as previous releases of the same passive tracer were conducted in smaller-scale convective flow systems and included a much smaller spatial and temporal separation of the release and sampling. Before we summarise the results we need to consider that such an experiment requires (i) a tracer gas with very low natural atmospheric concentrations; (ii) if a suitable and highly sensitive airborne sampling and analysis technique; (iii) accurate forecasts for

reliable flight planning; and (iv) a sophisticated flight planning method. We can report that increased values of the collected tracer gas coincided with sections of the Falcon flights where trajectory calculations suggested increased probabilities for the tracer air mass. However, the locations of the sampled tracer were characterized by rather low probabilities (i.e. they were not confirmed by many EDA members) and the tracer sampling just occurred at the edge of the WCB. Since we regard our technical setup for the experiment and the flight planning as sophisticated (Schäfler et al., 2014), we attribute the relatively

low tracer probabilities at the edge of the WCB – instead of the targeted high tracer probability in the core of the WCB – to a non-perfect forecast as mentioned in (iii) above, to obviously substantial uncertainties in the EDA, and finally to the limited area where the tracer gas was released into the WCB. The fact, however, that the observed transport of the tracer qualitatively



agrees well with the WCB pathway as indicated by EDA trajectories provides highly valuable evidence that WCBs, identified on the basis of trajectory calculations in many studies in the last 25 years, are meaningful Lagrangian flow features in the atmosphere.

WCB probabilities were also used to link the Falcon in situ measurements in space and time with earlier observations in the WCB, which we refer to as Lagrangian matches. This approach revealed that part of the WCB inflow was already observed at low levels by a HyMex lidar near Montpellier, and later at the southern slope of the Alps as the WCB passed a MeteoSwiss radar. These Lagrangian matches between in situ airborne and ground-based observations occurred within the WCB air as confirmed by the tracer. The Lagrangian matches were not planned beforehand, but were beneficially included later during the analysis of this WCB. EDAs are also used for evaluating humidity and cloud properties with the measurements and to characterise the evolution of the moisture in the model. The key findings from this evaluation can be summarised as follows:

– Lidar humidity measurements sampled both branches of the WCB inflow from the Atlantic and Mediterranean, respectively, during a time period of almost one day. Irrespective of the changing wind direction due to the approaching cyclone and the local wind circulation (Di Girolamo et al., 2009), the observations reveal an almost continuously too moist boundary layer in the ECMWF analyses. This highlights the difficulty to correctly represent moisture in this complicated flow situation of the WCB inflow, which was influenced by multi-scale processes. Our results also agree with earlier observations by Schäfler et al. (2011) and Schäfler and Harnisch (2015) who also found a too humid boundary layer in WCB inflows in analysis data.

– While the North Atlantic WCB branch already started to ascend, the Mediterranean branch moved further east taking up moisture along the coast before impinging upon the Alps. There, the MeteoSwiss radar observed intense surface rainfall and mid-level snow associated with the WCB. The particular WCB air parcels that were previously observed by the lidar and subsequently sampled by the Falcon are part of this precipitating cloud close to the melting layer. A few hours earlier, the radar observed WCB air from both the Mediterranean and the Atlantic branches and the two branches were vertically separated. Comparing the moisture and cloud formation along the WCB trajectories, the dryer Mediterranean branch almost instantaneously formed dense high-reaching clouds when it was forced to ascend at the Alps, while the clouds of the Atlantic branch developed more gradually and were less deep. The radar-observed WCB cloud in our case is rather stratiform and does not exhibit characteristics of embedded convection, as observed in other (non-orographic) WCB cases.

– Later during the further ascent of the WCB north of the Alps, the Falcon measured in situ water vapour and condensate in the mid and upper troposphere, including Lagrangian matches with the earlier observations. Several successful WCB intersections showed that water vapour in the cloudy WCB is often lower in the EDA compared with the measurements. In regions below the WCB, where according to the EDA snow falls out of the WCB and presumably sublimates, water vapour seems to be distinctly underestimated in the model. Cloud condensate in the WCB was found to be of similar magnitude in the EDA compared to observed cloud condensate in the mid-troposphere (mixed-phase cloud) and slightly





underestimated further above (ice cloud). To our knowledge, these are the first reported in situ measurements of cloud condensate in a WCB, repeated two years later as part of the ML-Cirrus aircraft campaign (Voigt et al., 2017).

Apart from this appealing results, it is important to emphasise the limitations of this campaign-driven study. The limitations of the study are related to the fact that even with a sophisticated flight planning, perfect flight routes in terms of maximising
WCB encounters and Lagrangian matches have not been possible for several reasons. As an effect, the number of Lagrangian matches between flights and the duration of in situ observations within the WCB is limited. We think that this is the prize to pay for such a challenging observational study that essentially relies on flight planning, which in turn was based on non-perfect forecasts. However, we think that this multi-faceted study reveals a range of interesting and, for some of them, unique features of WCBs. This study presented the first case of a WCB with measurements following the Lagrangian ascent and
describing the moisture and cloud evolution along the flow. The limited number of observations, however, only allowed to analyse a limited set of processes contributing to the total moisture budget of this WCB. We expect that repeating this kind of Lagrangian measurements in a WCB with a more complete instrumental package will be rewarding. In addition, the theme of orographic WCBs definitively deserves further attention as an orographcally induced change in updraft velocities may modify the dynamics and the water partitioning within the WCB.

*Author contributions.* MB designed this study, performed the trajectory calculations, and analyzed the data. AS directed the aircraft campaign, and MB, HSo, CV and HW contributed to the flight planning. SK and CV provided the measurements of water vapour and total water; HSch led the passive tracer experiment; DS and PDG the lidar observations, DN and UG the radar observations, and MS expertise about Alpine flow dynamics. MB and HW wrote the paper with input from all co-authors.

*Data availability.* The data are available from the authors upon request.

*Competing interests.* The authors declare that they have no conflict of interest.

*Acknowledgements.* We are most grateful for the scientific and financial support from Ulrich Schumann and Andreas Dörnbrack (both DLR), which made this aircraft campaign possible. HW acknowledges funding from ETH Zurich that supported the campaign. MB acknowledges funding by the Swiss National Science Foundation (grant no. 165941) and the European Research Council 485 (ERC) under the European Union's Horizon 2020 Research and Innovation program (project INTEXseas, grant agreement no. 787652). CV and SK acknowledge
funding by the Helmholtz Association under contract W2/W3-60 and by the German Research Foundation within DFG-SPP 1294 HALO (Vo1504/4-1) and DFG-SPP 2115 PROM (Vo1504/5-1). HSch acknowledges funding of the tracer work by the German Science Foundation (DFG) through the priority program SPP1294, project SCHL1857/1-2.





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
