# Peer review of "Lagrangian matches between observations from aircraft, lidar and radar in a warm conveyor belt crossing orography"

_Atmospheric Chemistry and Physics, 2020_

## Referee Comment (RC1) · Anonymous Referee #1 · 23 Nov 2020

The authors present the results of a unique and ambitious observational experiment aiming to sample air on its path within a warm conveyor belt airmass several times over the timescale of 1-2 days that the WCB exists. Although Lagrangian experiments have been conducted before, this is the first time to my knowledge that a deliberate tracer release has been used to test beyond doubt whether or not the same air is intercepted on later occasions over a day later in a WCB. In contrast with ground-based tracer release (such as ETEX in 1994) where tracer measurements later were sparse because the network of measurement sites were in the BL, but most of the tracer left the BL, the tracer release in T-NAWDEX-Falcon extended throughout the depth of the boundary layer (from a light aircraft) and the interception was made in the

upper troposphere using an aircraft directed to forecast interception locations. In the ITCT-Lagrangian 2004 experiment strong evidence that multiple air mass interceptions were made across the North Atlantic was established, but these were not verified using deliberate tracer release and so some degree of uncertainty remains.

The tracer interceptions occurred at times along the two downstream flights consistent with the calculated trajectories from the release flight track. An ensemble of analyses (from the ECMWF EDA) was used to account for uncertainty in the resolved wind fields used to calculate the trajectories. A complexity was that the calculated WCB trajectories from the small release area grouped into two coherent branches crossing the Alps in different locations, but the tracer measurements are consistent with the existence and path of these two branches. The difference in trajectories relates primarily to the altitude of release. Although the authors appear disappointed in conclusion that the WCB air masses tagged with tracer release were intercepted on the edges, I think it is remarkable that it was achieved at all in such a complex flow with multiple branches over a major mountain range with active precipitation and strong vertical motion. To my mind, it verifies that the horizontal paths and even vertical motion of trajectories calculated from analyses must have a close resemblance to the actual path of air. However, Fig.9b illustrates how air from a relatively small volume is strung out along a very long band in the WCB (in this case spanning across Germany from SW to NE). Since there must be very substantial dilution of the tracer through mixing in this environment with strong shear dispersion, it is also impressive that the detection limit is so low that the tracer can be measured and unambiguously attributed to the release.

The authors use the Lagrangian matches with air sampled above two ground-based profiling sites to examine the time history of water in all its phases along the WCB. This is the second major novel part of the investigation. It is found that water vapour is over-estimated in analyses within the BL at the origin of the WCB. However, the sum of simulated ice and liquid water content is consistent with observations on the flight track.

I recommend publication subject to minor revisions which clarify the Lagrangian connections on all the figures. Also, in the summary the authors refer to the difficulty with planning the experiment so that the tagged WCB air could be intercepted. A major uncertainty (I suspect the greatest uncertainty) relates to the use of forecast trajectories to direct the aircraft. The importance of this uncertainty could be estimated by showing the forecast trajectories from the release flight track (using the same lead time for forecast winds as used in conducting the experiment) and comparing them with trajectories calculated using analyses (as shown in Fig. 9b and 10). The consistency between the analysed Lagrangian match trajectories and tagged tracer measurements is very impressive – all the more so if the paths of forecast trajectories turn out to be less consistent with the measurements.

Specific comments and revisions

Section 1: Should be more precise on the novel aspects of this experiment. For example, tracer release experiments have been attempted over long-range (>1000 km) using release and a network of measurements within the boundary layer (e.g., ETEX in Van Dop et al, 1998, Atm. Env., 32, 4089-4094, doi.org/10.1016/S1352-2310(98)00248-9.). This section misses references to this work. However, I believe the T-NAWDEX-Falcon experiment is the first time that release and interception have been attempted using aircraft with a long time, distance and altitude difference between them.

l.47: Similarly, it is stated that "this is the first study that describes Lagrangian matches between humidity measurements in a WCB". However, this was done in the ITCT-Lagrangian experiment in terms of specific humidity (and theta-e) only, but not in terms of liquid and ice cloud condensate and without verification from deliberate tracer tagging. There are also quasi-Lagrangian experiments that have attempted to follow the evolution of clouds over shorter range (using airborne cloud microphysics measurements, e.g., ASTEX). Finally, there are aircraft experiments that have examined transport within WCBs using trace chemical measurements, but without the benefit of measurements in a Lagrangian frame. For example, Bethan et al, 1998, J. Geophys. Res., 1031(D11), 13413-13434, doi: 10.1029/98JD00535. A rather similar case in terms of a WCB running across France, Switzerland and Germany was examined in the EXPORT experiment (Purvis et al, 2003, J. Geophys. Res., 108(D7), 4224, doi:10.1029/2002JD002521). Some reference to these observational studies is required and their limitations compared to your approach in the T-NAWDEX-Falcon experiment.

Figures 4 and 5. Labelling the Lagrangian matches on figures. Although you do not introduce the notation until Section 4.2, it would be very useful to label the locations of matches T1 and T2 on these plots (especially the top panels) and then explain the triple "Lagrangian matches" later.

Figures 4 and 5. Relative error in Qv is not defined. I would have assumed $100*(Q\_EDA-Qv)/Qv$. However, in both figures it seems to vary about 100% and the 100% line is marked. So is the quantity shown actually $100*Q\_EDA/Qv$ ?

l.446: The physical picture associated with the radar section in Fig.8b is not clear to me. You point out the WCB trajectories (red dots) in the western half (west of 8.8E) and the region with less precip and non-WCB trajectories (blue dots) in the middle. However, why is there this region with less precipitation and what is the heavier precipitation east of 9.0E associated with? This is no longer the WCB air mass? Can you explain why these structures are there?

Fig. 8c: Many of the black contours in this figure stop in the middle. Why is that? I don't understand what they represent.

Figures 9 and 10: The 600 hPa ascent criterion used to label trajectories as WCB trajectories is having rather a large influence here, although it is arbitrary. You can see from the lower panel of Fig.9a that all forward trajs from the release track ascend a long way, but the majority less than 600 hPa. However, these ones start from a higher altitude since the release track goes up to 750 hPa. Indeed, it looks as though all the trajectories from this release track reach a pressure level of about 300 hPa and

the variation is Dp is mainly to do with the altitude of the release aircraft. Therefore it seems likely that the whole release track is within the WCB airmass and is all destined to reach a similar outflow level in 48 hours time. So the "WCB probabilities" shown in the Fig.10 cross-sections must naturally only identify the upper flank of the WCB since the ascent criterion is so strong. The fact that the "tracer probability" is high in gaps between the WCB probabilities is not especially relevant other than indicating that the tracer was released into air that travelled beneath the upper flank of the WCB (and therefore presumably nearer the middle of the air mass).

l.530: "technical issue concerning the manual time adjustment of the device cannot be completely excluded". The issue is not explained. Do you mean to say that the position of the samples on the time axis in Fig.10a is uncertain? If so, by how much? It looks to me that it cannot be much since the tracer is detected in the Mediterranean WCB airmass on the aircraft ascent and descent and also the first detection at 400 hPa and above coincides with Lagrangian match T1 and entry into the upper flank of the WCB. Also, the detection on Flight IOP2c is coincident with trajectory match T2. Surely this cannot be chance?

Fig.10: It would be good to label points T1 and T2 on the cross-sections to help connections back to earlier figures.

Fig.10 interpretation: After 08:54 (Lag match T1) the aircraft flew in the upper flanks of the WCB and back trajectories from the flight track went back to the Atlantic (Fig.6a). Despite this origin, tracer was detected. However, forward trajectories from the tracer release (Fig.9b) follow almost the same horizontal path over Germany just beneath the trajectories of Atlantic origin (Fig. 10a). This behaviour is seen clearly in Fig.2b where all "WCB trajectories" are shown. There seem to be two possible explanations that are not mutually exclusive:

A) The trajectories are calculated following the resolved, laminar flow represented by the analyses and the outflow layers are shallow with trajectories from different origins

coming very close. Vertical mixing by sub-grid scale motions would be expected and this maybe responsible for mixing the tracer upwards into the upper flank of the WCB (where the aircraft interception is).

B) Net ascent is under-estimated by the trajectory calculation using the analyses and the air of Med origin reaches a slightly higher level where the aircraft was flying.

Note that Lagrangian trajectory match T2 has an excellent match with the one elevated tracer sample on IOP2c. A similar vertical mixing argument to (A) was presented in Purvis et al (2003) to explain the measurements of short-lived hydrocarbons above the upper flank of the WCB calculated using trajectories. In that summer case, embedded convection in the WCB was important, giving rapid vertical mixing, while in your case the radar observations convincingly demonstrate that this WCB was not convective and vertical mixing would be expected to be slower.

This merits discussion in the conclusions. Given the very large shear dispersion of the tracer gas along the WCB (Fig.9b) and turbulent vertical mixing, it is very strong evidence that the trajectory calculations are a good representation of transport in the atmosphere since tracer was detected in the Med WCB on IOP2b and at T2 in IOP2c and was also detected in the upper flank of the WCB near T1 on IOP2b which must have been at the leading edge of the advancing tracer. One question that is not addressed is the dilution of the tracer. The release amount must be known (in kg) and it was distributed along a track (evenly?). It is detected with mixing ratio of the order $\sim$100 ppqv. So it must be possible to estimate approximately the volume of air containing the tracer (at 0900 15 Oct) and what this implies for the average depth of the tracer layer if the horizontal extent is given by the black dots in Fig.9b. Is this estimate consistent with the vertical range of the blue tracer probability in Fig.10? I think this would be important to deduce if mixing and explanation (A) above can account for the observations. It would be fascinating to know this average tracer depth estimate.

l.570: As argued above, the fact that the tracer probability maximum lies below the

"WCB probability" maximum is to be expected given the high threshold used on trajectory ascent to define WCB trajectories. So this does not indicate a failure of the experimental methodology. The tracer is in the WCB.

l.571: In the forecast methodology used for targeting the WCB an ascent criterion was used to isolate a subset of WCB trajectories. So, in order to distinguish the factors resulting in the greatest uncertainties it would be necessary to examine the forecast trajectories. I suggest you calculate forward trajectories from the release flight track using forecast winds (with the lead time used at the time) and compare the results with Fig.9b and Fig.10. Was the Falcon flight track above the maximum tracer probability obtained from forecasts (as it is using analyses)? Is the mismatch associated with forecasting the winds?

Technical corrections

Title: I am not sure that "orographic warm conveyor belt" is accepted terminology. I suggest, "Lagrangian matches between observations from aircraft, lidar and radar in a warm conveyor belt crossing orography"

l.6: "wind fields of the ECMWF ensemble data assimilation system were used" is not specific enough. I suggest "an ensemble of wind fields from the global analyses produced by the ECMWF Ensemble Data Assimilation (EDA) system".

Fig.2: The panels are small with a lot of white space around them. I looks like 2x2 panels should work but please expand figure panels to fill the page column width.

Fig.3: There are a lot of details in the panels, but much too small to see (especially panel b with the flight tracks overlain). I am not convinced that panels a and c are needed. I think it would be better to present only panel b, much larger with key locations of ground stations and flight tracks marked.

l.407: "ascended much further WEST compared to the rest of the WCB"?

l.428: Correction: Should be referring to Fig. 8a (not Fig. 7a).

l.431: the yellow hatching is on Fig. 8a.

l.438 and Fig.8b: The red asterisk associated with T1 is very hard to spot. Please label this "T1" within the figure panel. Similarly, label the red asterisk associated with T2 in Fig. 8c.

l.456: Trajectory crosses Montpellier (LEFT grey bar in the middle panel of Fig. 7b).

Fig.7: The colours used for IWC and RWC are both blue and similar. They can be distinguished in the cross-sections but it is hard to tell them apart in the Qc graphs. Perhaps change one to a more distinct colour?

l.512: "In the evening of 14 October"

---

## Referee Comment (RC2) · Josué Gehring (Referee) · 26 Nov 2020

**1 General comment**

The authors investigate an orographic WCB with a very innovative Lagrangian matching involving ensemble WCB trajectories and in-situ measurements of a tracer released in the inflow of the WCB. This study presents at least three major novel aspects. The first one is the method: to my knowledge, it is the first time that Lagrangian trajectory computation of a WCB could be verified by measuring in the outflow region a tracer gas that was released in the inflow. This must have been a very challenging task involving accurate ensemble trajectory predictions and strategic flight planning. More importantly, it shows that trajectory computation are reliable enough to plan in-situ aircraft measurements. The second aspect is the triple Lagrangian matching: to have measurements in the inflow, ascent and outflow of a WCB allows to better characterise the humidity conditions in these three different phases of the WCB and to evaluate the model performances along them. This is a crucial contribution for future studies on model verification. The third one is the orographic aspect of the WCB "T2". To my knowledge there has not been many studies mentioning the orographic contribution to the WCB ascent. This a nice example of a WCB ascent with significant orographic lifting. For these reasons, I strongly support the conclusions of this study and its publication in Atmospheric Chemistry and Physics, subject to minor revisions.

**2 Minor comments**

- P.6, L.178-179: could you briefly justify the choice of -30 C for the threshold between saturation with respect to liquid water and ice ?

- P.7, L. 201: "Only data outgoing from the instrument upward until a relative error of 100% is reached are used, [...]". I am not sure to understand this sentence, it means you compute the relative error with respect to the radiosounding measurement and when it reaches an altitude where it is greater than 100% you take this altitude as the maximum measurement height for this period? Do you think the horizontal displacement of the radiosounding could significantly contribute to this relative error (and maybe also to the calibration)?

- P.9, L.250: did you consider using the modification of this condition proposed by Binder2020 et al. 2020 (i.e. consider also trajectories that fulfil the 600 hPa ascent before 48h, but then descend)? Do you think that could impact the result

of this study? Maybe your WCB selection was performed before the study of Binder et al. 2020, in which case I totally understand that it was not used.

- P.10, L.294: "On a smaller scale, the strong westerly wind ahead of the cold front initiates a secondary lee cyclone [...]". Which figure and region are you referring to with this westerly wind? Do you mean the westerly wind over Spain in Fig. 2c? In this case the Piemont region is not in the lee of this flow. Or you mean the westerly wind over France in Fig. 2a or 3b? In this case it is not ahead of the cold front.

- P.15, L.363: "[...] facilitates aggregation and riming[...]". Supercooled liquid water favours riming, but not necessarily aggregation. It was the wind shear and turbulence below the WCB air masses, which promoted aggregation in Gehring et al. 2020.

- P.15, L. 384: "[...] is likely to sublimate or evaporate." Do you have an idea why sublimation is not simulated in ERA5?

- P.18, L.432: "[...] if the model is assumed to underestimate precipitation from the WCB [...]". Does the model underestimate precipitation from the WCB?

- P.19, L.461: "The moist bias in EDA does not seem to depend on the large-scale flow direction." Do you have an hypothesis on the reason for this moist bias in EDA?

- P.21, L.490: "[...] dynamically-driven ascent for T1 [...]". Do you know if the orography also significantly influenced the ascent of T1?

- P. 21, L.482: "[...] the orographic lifting coincided most likely with enhanced dynamical forcing for ascent [...]". The respective contribution of orographic and dynamic lifting in the ascent of the WCB is very interesting. Out of curiosity: do

you think it is possible to disentangle both effects to quantify what is the contribution from each? If yes, how would you suggest to do it?

- P.24, L.430-431: "Also, a technical issue concerning the manual time adjustment of the device cannot be completely excluded." From what you wrote on P.8, L.232-233 it seems that this issue was almost confirmed. Here it sounds as if it would be a possible issue that might have taken place. If this issue is likely as you mention on P.8, maybe it would be worth rephrasing this sentence and referring to P.8 (e.g. a technical issue [...] likely occurred as mentioned in Sect. 2.4). This would make it clear to the reader that it is an issue you likely had and not only a potential problem that you list among other possibilities explaining the time shift.

- P.27, L.617: "[...] with a more complete instrumental package [...]". If you could redo this campaign, which instruments would you add?

**3 Technical corrections**

- P.2, L.48: "[...] an unique airborne tracer [...]". Remove the "n" in "an".

- P.2, L. 84: "Oertel et al. (in preparation)". I think this article is in review in WCD discussion, so it would be better to cite it as such.

- P.3, L.87: "This brief summary reveals the importance of WCBs for understanding precipitation in and the dynamics of extratropical cyclones, [...]". Remove the "in".

- P. 10, L.284: "Within the westerly flow, WCB trajectories originating over the Atlantic are embedded that start to ascend prior to passing the lidar (Fig. 2b)." I do not understand this sentence, do you mean "[...] and start to ascend [...]"?

- P.15, L.366 "[...] (see blue line in Fig. 3a) [...]". You probably mean Fig. 3b.

- P.18, L.428 "[...] (Fig. 7a, between 16:00-19:00 UTC)." You probably mean Fig. 8a.

- P.19, L.475: "[...] (read and blue dots in Fig. 8c)". It seems that there are only asterisks on Fig. 8b,c, instead of asterisk for T2 and dots for WCB or non-WCB air. This makes it hard to read when you are referring to a single asterisk in the location of T2.

- P.21, L. 485: "[...] when the aircraft enters the region of high WCB probabilities (Figs. 5)". It should be Fig. 5 without "s".

- P.22, caption Fig. 8: add "N" after "46.15".

- P.22, Fig. 8c: the labels of the WCB probability are not very clear.

- P.25, L. 564: "[...] (ii) if a suitable [...]". Remove "if".

**4  References**

- Binder, Hanin, Maxi Boettcher, Hanna Joos, Michael Sprenger, and Heini Wernli. "Vertical Cloud Structure of Warm Conveyor Belts – a Comparison and Evaluation of ERA5 Reanalysis, CloudSat and CALIPSO Data." Weather and Climate Dynamics 1, no. 2 (October 19, 2020): 577–95. https://doi.org/10.5194/wcd-1-577-2020.

- Gehring, Josué, Annika Oertel, Étienne Vignon, Nicolas Jullien, Nikola Besic, and Alexis Berne. "Microphysics and Dynamics of Snowfall Associated with a Warm Conveyor Belt over Korea." Atmospheric Chemistry and Physics 20, no. 12 (June 25, 2020): 7373–92. https://doi.org/10.5194/acp-20-7373-2020.

---

## Author Comment (AC1) · 4 Feb 2021

We thank both reviewers for their effort to read the manuscript carefully. Their critical questions and constructive comments helped to further improve the quality of the paper.

Our document containing the detailed answers is uploaded as the supplement.

Please also note the supplement to this comment:
https://acp.copernicus.org/preprints/acp-2020-1019/acp-2020-1019-AC1-supplement.pdf

[Figure]

[Figure]

**Supplement:**

**Paper acp-2020-1019**

**Lagrangian matches between observations from aircraft, lidar and radar in an orographic warm conveyor belt**

**by M. Boettcher et al.**

**Response to the reviewers' comments**

We thank both reviewers for their effort to read the manuscript carefully. Their critical questions and constructive comments helped to further improve the quality of the paper.

**Reviewer #1:**

**1.** *Section 1: Should be more precise on the novel aspects of this experiment. For example, tracer release experiments have been attempted over long-range (>1000 km) using release and a network of measurements within the boundary layer (e.g., ETEX in Van Dop et al, 1998, Atm. Env., 32, 4089-4094, doi.org/10.1016/S1352-2310(98)00248-9.). This section misses references to this work. However, I believe the T-NAWDEX-Falcon experiment is the first time that release and interception have been attempted using aircraft with a long time, distance and altitude difference between them.*

Reply:
We are very thankful for the detailed literature suggestions about the past tracer experiment ETEX, which we missed to mention in the introduction. We also adapt our statement about the novelty of our study and expand our introduction by discussing the former literature as follows (see also next comment)

2$^{nd}$ paragraph of the introduction:
"To the best of our knowledge, this is the first successful study that describes Lagrangian matches between aircraft observations in a WCB, confirmed by a tracer release experiment with a large horizontal and vertical separation between release and intercept. The unique airborne tracer release experiment was performed as part of T-NAWDEX-Falcon to provide direct experimental evidence for the long-range transport by the WCB."

**2.** *l.47: Similarly, it is stated that "this is the first study that describes Lagrangian matches between humidity measurements in a WCB". However, this was done in the ITCTLagrangian experiment in terms of specific humidity (and theta-e) only, but not in terms of liquid and ice cloud condensate and without verification from deliberate tracer tagging. There are also quasi-Lagrangian experiments that have attempted to follow the evolution of clouds over shorter range (using airborne cloud microphysics measurements, e.g., ASTEX). Finally, there are aircraft experiments that have examined transport within WCBs using trace chemical measurements, but without the benefit*

*of measurements in a Lagrangian frame. For example, Bethan et al, 1998, J. Geo-phys. Res., 1031(D11), 13413-13434, doi: 10.1029/98JD00535. A rather similar case in terms of a WCB running across France, Switzerland and Germany was examined in the EXPORT experiment (Purvis et al, 2003, J. Geophys. Res., 108(D7), 4224, doi:10.1029/2002JD002521). Some reference to these observational studies is required and their limitations compared to your approach in the T-NAWDEX-Falcon experiment.*

Reply:
Many thanks for critically pointing to our statement about the novelty aspects. We corrected this statement as written above. In addition, we highly appreciate your references for past Lagrangian and WCB experiments and revised the 7[th] paragraph of the introduction as follows:

"In principle, such Lagrangian matches enable investigating the material evolution of thermodynamic variables along a WCB. A major challenge of such experiments is the planning of the Lagrangian matches realised by aircraft. In contrast to the boundary layer experiment ETEX in 1992 where additional balloons were used to follow the mainly 2-dimensional movement of trade wind clouds (e.g. Bretherton and Pincus, 1995; Sigg and Svensson, 2004), the prediction of the ascending WCB relies on air parcel trajectories using 3-dimensional wind fields from forecasts, which are inherently uncertain. To cope with this uncertainty, the planning of Lagrangian matches is best done with data from ensemble forecasts (Schäfler et al., 2014; 2018). An interesting observational approach to identify Lagrangian matches is the use of a physical tracer that is measured at consecutive times to experimentally corroborate the pathway of air parcels. Former experiments have proven that naturally occurring boundary layer trace gases and pollutants can effectively be transported by the WCB to upper levels of the free troposphere (Bethan et al., 1998; Purvis et al., 2003). Using this property, an experiment in 2004 described in Methven et al. (2006) aimed at realising Lagrangian matches between airborne measurements in the free troposphere to study intercontinental transport of pollutants. One case of the campaign involved a WCB, for which the natural occurrence of a physical tracer was used to mark air parcels. Another extensive experiment over Europe in 1994 simulated an emergency situation by the release of perfluorocarbon tracers in the planetary boundary layer (Van Dop et al., 1998). Their second tracer release, lasting 12 hours, involved a WCB-like situation of a cold front passage and it was reported that an aircraft sampled the tracer 700–800 km away from the source (Nodop et al., 1998). The results of these upper air samples, however, do not appear in the literature and most likely failed. The approach in our study is, for the first time, to specifically focus on studying the transport along a WCB by the release and re-sampling of a synthetic inert tracer (Ren et al., 2015). For completeness, we briefly note that Lagrangian matches have also been applied in research on stratospheric chemistry, for instance, by Rex et al. (1998) to infer ozone loss rates in the Arctic stratosphere from ozonesonde measurements, and by Fueglistaler et al. (2002) to study the Lagrangian evolution of polar stratospheric clouds from consecutive airborne lidar observations."

**3.** *Figures 4 and 5. Labelling the Lagrangian matches on figures. Although you do not introduce the notation until Section 4.2, it would be very useful to label the locations of matches T1 and T2 on these plots (especially the top panels) and then explain the triple "Lagrangian matches" later.*

Reply:
You are right, showing the intersection of the matching trajectories would make them more prominent. We decided, however, to only show the intersections of T1 and T2 because showing all of them would make the plot very busy. We include markers for T1 and T2 in Figs. 4, 5, 8 and 10.

**4.** *Figures 4 and 5. Relative error in Qv is not defined. I would have assumed 100\*(Q_EDA-Qv)/Qv. However, in both figures it seems to vary about 100% and the 100% line is marked. So is the quantity shown actually 100\*Q_EDA/Qv ?*

Reply:
The formula we have used is indeed the second one you mention. We changed Figs. 4 and 5 and now apply your first and more common equation. To make it clear, we write the formula in the caption.

**5.** *l.446: The physical picture associated with the radar section in Fig. 8b is not clear to me. You point out the WCB trajectories (red dots) in the western half (west of 8.8E) and the region with less precip and non-WCB trajectories (blue dots) in the middle. However, why is there this region with less precipitation and what is the heavier precipitation east of 9.0E associated with? This is no longer the WCB air mass? Can you explain why these structures are there?*

Reply:
One reason for the rather complex structures in radar reflectivity is that the two branches of the WCB both intersect the radar cross section at the same time. We therefore have redone Fig. 8b in the manuscript, but now separately for the WCB branches with inflow from the North Atlantic and the Mediterranean ( Fig. R1).

One can now see that the highest reflectivities caused by the melting hydrometeors in the lowest levels consists of Mediterranean WCB with up to 100% probability (Fig. R1 right). The North Atlantic WCB is visible above and mainly in the western part of the radar cross section with probabilities of more than 30%. At p ~ 600 hPa, WCB probability from the Atlantic decreases from west to east. Hence, the low reflectivities in the middle of the radar cross section coincide with Atlantic WCB probabilities between 10 and 1%. Most likely, this air started raining out already before impinging upon the mountains. Further east, reflectivities increase again but WCB probability goes down to zero. This air east of 9.2°E and at p <= 600 hPa originates from the North Atlantic and does not fulfil the WCB criterion. The air partly arrives from slightly elevated levels ahead of the WCB. We cannot explain why this air caused precip with slightly higher radar reflectivities than the WCB air further west.
We revise the description of the radar image in Fig. 8b in the manuscript and add a few more details.

[Figure]

**Fig. R1:** Vertical cross-sections of radar reflectivity from the Monte Lema radar (colours, dBZ), and intersection points of Falcon-probed trajectories (blue) and Falcon-probed WCB trajectories (red), at 23:40 UTC 14 October with Lagrangian matches with flight IOP2b. The black contours denote WCB probabilities with inflow from the North Atlantic (left) and the Mediterranean (right). The red dot marks the intersection of T1. The combination of the two panels corresponds to Fig. 8b in the manuscript.

**6.** Fig. 8c: *Many of the black contours in this figure stop in the middle. Why is that? I don't understand what they represent.*

Reply:
We apologise for the poor quality of the lines in Figs. 8b and c, they denote WCB probability. We managed to make them clearer.

**7.** *Figures 9 and 10: The 600 hPa ascent criterion used to label trajectories as WCB trajectories is having rather a large influence here, although it is arbitrary. You can see from the lower panel of Fig. 9a that all forward trajs from the release track ascend a long way, but the majority less than 600 hPa. However, these ones start from a higher altitude since the release track goes up to 750 hPa. Indeed, it looks as though all the trajectories from this release track reach a pressure level of about 300 hPa and the variation in Dp is mainly to do with the altitude of the release aircraft. Therefore it seems likely that the whole release track is within the WCB airmass and is all destined to reach a similar outflow level in 48 hours time. So the "WCB probabilities" shown in the Fig. 10 cross-sections must naturally only identify the upper flank of the WCB since the ascent criterion is so strong. The fact that the "tracer probability" is high in gaps between the WCB probabilities is not especially relevant other than indicating that the tracer was released into air that travelled beneath the upper flank of the WCB (and therefore presumably nearer the middle of the air mass).*

Reply:
The trajectories from the tracer release in Fig. 9a shown in the Dp panel are calculated forward and backward (same as for the trajectories in Figs. 4 and 5). Then, we apply the WCB criterion along each 48 h time interval along the trajectory. If the criterion is fulfilled for an 48-h time interval that includes the time of the flight, then the trajectory is labelled as WCB. This means that trajectories have the chance to fulfil our strict WCB criterion even though they are already at higher altitudes at the time of the tracer release shown in Fig. 9a. Figure 9a therefore  shows that the tracer moved within WCB air, but we were not able to

catch the strongest ascending part of the WCB with our measurements. See also reply to question 11. Unfortunately, we did not understand the last part of the comment about the upper flank of the WCB.

**8.** *l.530: "technical issue concerning the manual time adjustment of the device cannot be completely excluded". The issue is not explained. Do you mean to say that the position of the samples on the time axis in Fig. 10a is uncertain? If so, by how much? It looks to me that it cannot be much since the tracer is detected in the Mediterranean WCB air mass on the aircraft ascent and descent and also the first detection at 400 hPa and above coincides with Lagrangian match T1 and entry into the upper flank of the WCB. Also, the detection on Flight IOP2c is coincident with trajectory match T2. Surely this cannot be chance?*

Reply:
We discussed the results of the tracer probes with the instrument operators extensively in the past. For IOP2b we noticed an apparent time shift between the tracer measurements and Flexpart dispersion calculations, which we do not show in the paper. The time shift is also visible by the tracer probability from EDA trajectories in Fig. 10a. Our rough estimate would be a shift of about 20 minutes. Possible causes for the timing discrepancy might be of atmospheric nature by convection or turbulent mixing, or due to a technical issue during the sampling or the analysing procedure. For the latter, one conceivable explanation could be a a 20 min offset in the manual time adjustment for the sampling procedure. However, we have no actual evidence regarding this potential error source. For IOP 2c, tracer measurements and the dispersion simulation agree better and do not show any time discrepancy, which argues for natural displacement of the tracer gas for IOP2c rather than a (systematic) technical issue. While a time offset in the sampling procedures during flight IOP2b can ultimately not be entirely excluded, we have no indication that this is a more likely error source than differences between the analysis data and the actual flow, such as small-scale convection and unresolved windshear. We slightly change the text in section 4.3 follows:

"The measurements, however, do not exactly agree in time with the proposed peaks in tracer probabilities from the trajectories. The matching (and estimated tracer transport) based on kinematic trajectories comes with some uncertainties. For instance, mixing processes that, e.g., by deep convection in the WCB inflow and later over the Baltic Sea (see Figs. 2d, 3a), might have led to significant dispersion and possibly to a slight shift of the long-range transport in the atmosphere compared to the trajectories, which do not contain the effects of local-scale dispersion. Also, a technical issue concerning the manual time adjustment of the device cannot be completely excluded. When taking a roughly estimated the time shift of 20 minutes into account, the tracer measurements and probabilities would show a better agreement."

**9.** *Fig.10: It would be good to label points T1 and T2 on the cross-sections to help connections back to earlier figures.*

Reply:
Good point, done.

**10.** *Fig. 10 interpretation: After 08:54 (Lag match T1) the aircraft flew in the upper flanks of the WCB and back trajectories from the flight track went back to the Atlantic (Fig. 6a). Despite this origin, tracer was detected. However, forward trajectories from the tracer release (Fig. 9b) follow almost the same horizontal path over Germany just beneath the trajectories of Atlantic origin (Fig. 10a). This behaviour is seen clearly in Fig. 2b where all "WCB trajectories" are shown. There seem to be two possible explanations that are not mutually exclusive:*

*A) The trajectories are calculated following the resolved, laminar flow represented by the analyses and the outflow layers are shallow with trajectories from different origins coming very close. Vertical mixing by sub-grid scale motions would be expected and this is maybe responsible for mixing the tracer upwards into the upper flank of the WCB (where the aircraft interception is).*

*B) Net ascent is under-estimated by the trajectory calculation using the analyses and the air of Med origin reaches a slightly higher level where the aircraft was flying.*

*Note that Lagrangian trajectory match T2 has an excellent match with the one elevated tracer sample on IOP2c. A similar vertical mixing argument to (A) was presented in Purvis et al. (2003) to explain the measurements of short-lived hydrocarbons above the upper flank of the WCB calculated using trajectories. In that summer case, embedded convection in the WCB was important, giving rapid vertical mixing, while in your case the radar observations convincingly demonstrate that this WCB was not convective and vertical mixing would be expected to be slower.*

*This merits discussion in the conclusions. Given the very large shear dispersion of the tracer gas along the WCB (Fig. 9b) and turbulent vertical mixing, it is very strong evidence that the trajectory calculations are a good representation of transport in the atmosphere since tracer was detected in the Med WCB on IOP2b and at T2 in IOP2c and was also detected in the upper flank of the WCB near T1 on IOP2b which must have been at the leading edge of the advancing tracer.*

*One question that is not addressed is the dilution of the tracer. The release amount must be known (in kg) and it was distributed along a track (evenly?). It is detected with mixing ratio of the order about100 ppqv. So it must be possible to estimate approximately the volume of air containing the tracer (at 0900 15 Oct) and what this implies for the average depth of the tracer layer if the horizontal extent is given by the black dots in Fig. 9b. Is this estimate consistent with the vertical range of the blue tracer probability in Fig. 10? I think this would be important to deduce if mixing and explanation (A) above can account for the observations. It would be fascinating to know this average tracer depth estimate.*

Reply:
We agree with your observation that some kind of mixing of the Med and Atl WCB must have occurred in order to explain the tracer measurements of IOP2b between 08:55 and 09:30 UTC presented in Fig. 10. The tracer detection occurred around the northeasternmost part of the flight IOP2b (Fig. 2b). According to Fig. 6a, the WCB air that was observed in this region originates from the Mediterranean. Just a bit further west on the east-west flight leg, WCB trajectories at slightly higher altitude arrived from the Atlantic. First of all, we want to remind that a time shift of the tracer measurement compared to simulations is already discussed in the reply to question 8 above (and as stated in sections 2.4 and 4.3 in the manuscript). This would lead to an onset of the tracer measurements about 20 min later.

When we apply the same WCB separation into Atl and Med branches as in the reply to question 5 above, we can see that the tracer detection from 08:55 to 09:30 UTC overlaps with probabilities from both WCB branches (Fig. R2). This would hint to a mixing of the Atl WCB and the tracer (which is transported with the Med WCB), which most probably occurred due to turbulence as a result of wind shear at the front (see the strongly sheared trajectory positions in Fig. 9b). In addition, convection in the WCB inflow (Fig. 2d) and over the Baltic Sea (Fig. 3a) could have contributed to an enhanced mixing and dilution of the tracer. Figure R2 also reveals that when considering a potential time shift of the tracer measurements (as discussed in sections 2.4 and 4.3 in the manuscript and in the reply to question 8), the tracer detection is "pushed" towards higher WCB probabilities from the Mediterranean and also to higher trajectory-calculated tracer probability. Tracer probability and Med WCB clearly overlap in the cold front that is intersected by Falcon in its upper part at 09:30 UTC (Fig. R2 right). A time shift of the tracer measurements would shift the high values of the tracer gas towards this region.

[Figure]

**Fig. R2:** Tracer probability and measurements along Falcon flight IOP2b as in Fig. 10a of the manuscript. Upper panels show tracer probabilities (colours for all tracer trajectories, white hatching for tracer WCB trajectories), flight altitude (black line). Black contours show WCB probabilities (left) for the Atlantic WCB branch and (right) for the Mediterranean WCB branch (values of 1, 10, 30, 60 and 90%), and the lower panels show the tracer concentrations sampled onboard the Falcon. The red dot in the upper panel shows the intersection of the WCB trajectory T1.

In a back-of-the-envelope estimate, we calculate the initial tracer concentration after release, and the approximate dilution until the airmass is encountered by the aircraft. The tracer was released in an air volume of approximately 20x30 km, and over a pressure interval of about 200 hPa, which corresponds approximately to an airmass of 1e12 kg. According to aircraft records, 9.2 kg of the PMCP tracer (molar mass 300 g mol-1) were released. Using a molar mass of air of 28.9 kg mol-1, and assuming complete mixing in the release volume, one obtains a molar mixing ratio of 700 ppq in the airmass. Given the release flight conditions, these assumptions are most likely not fulfilled, and local concentrations will likely be substantially larger. As peak mixing ratios were on the order of 100 ppq, an at least 7-fold dilution of the airmass should be expected. However, the uncertainty of this ballpark estimate can easily exceed an order of magnitude, most likely in the direction that air was initially less diluted, and received dilution of up to another order of magnitude in the horizontal dimension. This interpretation would be in agreement with the strong horizontal shear apparent

in the trajectory calculations (Fig. 9b). The vertical dilution of the airmass (from 200 hPa to 300 hPa) is most likely secondary to the horizontal dilution of the airmass.

Because of the possible timing discrepancy between tracer measurements and model simulation (see reply to question 8) we decided to consider the measurements from a more qualitative perspective and would rather not include the theoretical quantitative estimation of the tracer concentration in the paper.

We already mention convection in the WCB inflow over the Mediterranean in section 4.3 in the manuscript as a possible source for dispersion and displacement of the tracer gas. We extend the discussion about turbulent mixing in the conclusions as follows: "What we cannot evaluate by using trajectories is dispersion or dilution of the tracer gas that possibly occurred by convective mixing in the WCB inflow and at the sampling location near the Baltic sea for IOP2b and also by turbulence due to wind shear along the cold front. These processes may have occurred in reality, potentially leading to a more widespread tracer plume as shown by our trajectories."

**11.** *As argued above, the fact that the tracer probability maximum lies below the "WCB probability" maximum is to be expected given the high threshold used on trajectory ascent to define WCB trajectories. So this does not indicate a failure of the experimental methodology. The tracer is in the WCB.*

Reply:
We are quite sure that catching the tracer within WCB air mass instead of within less ascending air was a matter of the flight timing and location. Fig. 9a in the manuscript shows that the tracer release occurred within very high (but not 100%) WCB probability. The resulting 58 tracer-carrying WCB trajectories (started every 1 min in 11 EDA members during the 20-min tracer release), however, are strongly sheared at the time of the flights as indicated by the black dots in Fig. 9b. As a consequence, it would not have been possible to reach a high tracer WCB probability even if the flight met the exact pathway of the tracer. But we agree, our WCB criterion is very strict and one can assume that the tracer-probed air as a whole was more or less within the WCB when applying a softer WCB criterion.

Inspecting Figs. R3 and R5 below, which show tracer paths in different forecasts from the flight planning, the actual way of the tracer in the EDA (Fig. 9b) was initially further west than expected from the forecasts. Only in the northern part over Germany and Poland (9:00 to 9:40 UTC in Fig. 10a) the forecasts better predicted the correct location to meet the tracer-carrying WCB. Therefore, the tracer sampling could have been (even) more successful if the flight sampled the WCB in its centre further west.

**12.** *l.571: In the forecast methodology used for targeting the WCB an ascent criterion was used to isolate a subset of WCB trajectories. So, in order to distinguish the factors resulting in the greatest uncertainties it would be necessary to examine the forecast trajectories. I suggest you calculate forward trajectories from the release flight track using forecast winds (with the lead time used at the time) and compare the results with Fig. 9b and Fig. 10. Was the Falcon flight track above the maximum tracer probability obtained from forecasts (as it is using analyses)? Is the mismatch associated with*

*forecasting the winds?*

Reply:

We did not use WCB probabilities from the ECMWF ensemble forecasts for the flight planning of the Lagrangian matches, we only used them to indicate the overall WCB occurrence (see Schäfler et al. 2014). We planned the tracer sampling flights only by calculating trajectories in the deterministic forecasts and with Flexpart dispersion calculations. In addition, we could not plan the sampling flights based on the latest and possibly most accurate forecast for the following reasons:

Due to the strong restrictions in the European airspace, we had to plan and submit the tracer release experiment very early. Hence, the forecast from 00 UTC 12 October 2012 served as the basis for the release and the sampling. Based on this forecast we saw that with a release at 12 UTC 14 October the tracer should be sampled 24h later over Germany (Fig. R3).

[Figure]

**Fig. R3:** Original figures from the aircraft campaign from forecasts initialized at 00 UTC 12 October 2012 that served for the planning of the tracer experiment. Left: Trajectories released at 12 UTC 14 October from the approximate region of the tracer flight with black markers at 12 UTC 15 October. The thin dark green line in the left panel marks the position of the vertical cross section shown in the right panel.

This forecast resulted in a concrete plan for the flights for IOP2 as shown in Fig. R4 including a tracer release and three sampling flights. For sampling flight IOP2a we realised later that the recording device was broken; therefore no results from flight IOP2a can be shown in the paper.

[Figure]

**Fig. R4:** preliminary flight planning for IOP2.

Since we were more flexible with the flight planning for the tracer release, we later moved the release forward to 09 UTC 14 October. Eventually, the forecast from 12 UTC 13 October led to a much faster (tracer) transport with the WCB despite the earlier scheduled tracer flight (compare Figs. R3 and R5).

[Figure]

**Fig. R5:** Same as Fig. R3, here for forecast base time 12 UTC 13 October and tracer release and trajectory start at 09 UTC 14 October.

As a consequence, we could manage to swap the flight routes of IOP2b and IOP2c on short notice to still have a chance to catch the tracer on IOP2b over northern Germany and Poland in the morning of 15 October. For this reason the tracer sampling flight route(s) and time(s) were not as exact as we had hoped in the beginning. IOP2c was eventually conducted over mid Germany in the afternoon that day. We did not even expect to sample

tracer on this flight and the (small) tracer concentrations observed (Fig. 10b) occurred rather "by accident". These explanations clearly illustrate the challenges in performing Lagrangian matches in a situation with very limited flight route flexibility, long flight planning lead times, and uncertain forecasts.

Thus, we prefer to not repeat the tracer probability analysis with, e.g., the ensemble forecast from 00 UTC 12 October 2012 because it does not reflect our flight planning procedure.

**Technical corrections**

**13.** *Title: I am not sure that "orographic warm conveyor belt" is accepted terminology. I suggest, "Lagrangian matches between observations from aircraft, lidar and radar in a warm conveyor belt crossing orography"*

Reply:
Thanks for the hint, we change the title.

**14.** *l.6: "wind fields of the ECMWF ensemble data assimilation system were used" is not specific enough. I suggest "an ensemble of wind fields from the global analyses produced by the ECMWF Ensemble Data Assimilation (EDA) system".*

Reply:
Thank you, we use the suggested formulation.

**15.** *Fig.2: The panels are small with a lot of white space around them. I looks like 2x2 panels should work but please expand figure panels to fill the page column width.*

Reply:
We increased the size of the panels to have less space in between them.

**16.** *Fig.3: There are a lot of details in the panels, but much too small to see (especially panel b with the flight tracks overlain). I am not convinced that panels a and c are needed. I think it would be better to present only panel b, much larger with key locations of ground stations and flight tracks marked.*

Reply:
We agree that panel c) is not necessary, but we prefer to keep the satellite image in panel a). We find the satellite image meaningful because it indicates the difficulty of hitting a moderately developed cloud band with an aircraft and that deep convection occurred ahead of the cold front near the coast of the Baltic Sea. The latter is also important for the discussion of the dispersion of the tracer gas. We add some references to the figure and more text to give the satellite image more importance:

p.13, l. 329:
"The morning flight IOP2b on 15 October intended to follow the gradual mid-level to upper-level ascent of the WCB from southern Germany towards the Baltic Sea as indicated by the WCB's cloud band in Fig. 3a."

p. 13, l. 358-360:
"At the level of the aircraft the maximum ascent of the observed air mass is highly uncertain and varies between 300 and 700 hPa in 48 h (Fig. 4c). There, the flight crosses the cold front once again and deep convection just ahead of the cold frontal cloud band near the Baltic Sea coast is visible in the satellite image (Fig. 3a)."

p. 24, l. 538ff:
"The measurements corroborate the presence of tracer gas with concentrations of up to 150 ppbv over a longer section along the flight. Exactly there, the maximum ascent of the trajectories started from the flight track was highly uncertain (Fig. 4c) and the satellite image indicates deep convection near the frontal cloud band (Fig. 3a). "

In addition, we increased the two panels of Fig. 3 in size and particularly tried to better highlight the marker and flight routes in b).

**17.** *l.407: "ascended much further WEST compared to the rest of the WCB"?*

Reply:
We changed the sentence to "*ascended much further northwest compared to the rest of the WCB".*

**18.** *l.428: Correction: Should be referring to Fig. 8a (not Fig. 7a).*

Reply:
Thanks, we changed it.

**19.** *l.431: the yellow hatching is on Fig. 8a.*

Reply:
You are right, we changed the reference.

**20.** *l.438 and Fig.8b: The red asterisk associated with T1 is very hard to spot. Please label this "T1" within the figure panel. Similarly, label the red asterisk associated with T2 in Fig. 8c.*

Reply:
We marked T1 and T2 in the respective panels with a distinctive red dot to make them better visible.

**21.** *l.456: Trajectory crosses Montpellier (LEFT grey bar in the middle panel of Fig. 7b).*

Reply:

Thank you, we added "left".

**22.** *Fig.7: The colours used for IWC and RWC are both blue and similar. They can be distinguished in the cross-sections but it is hard to tell them apart in the Qc graphs. Perhaps change one to a more distinct colour?*

Reply:
We changed the colour of RWC from purple to violet and made the lines in the lower panel thicker.  They should be easier to distinguish now.

**23.** *l.512: "In the evening of 14 October"*

Reply:
Thank you! Done.

**Reviewer #2 (Josué Gehring):**

**Minor comments**

**1.** *P.6, L.178-179: could you briefly justify the choice of -30 C for the threshold between saturation with respect to liquid water and ice ?*

Reply:
Since the ascending air masses originate in the boundary layer, we assume a mainly heterogeneous freezing behaviour of the cloud. In contrast to the homogeneous freezing threshold at ~-38°C, heterogeneous freezing already starts at higher temperatures while the exact freezing temperatures are strongly dependent on the CCN properties. Here we use -30°C as an estimated threshold where we would expect a rather high heterogeneous freezing nucleation rate (e.g. from Figure 6 in Khvorostyanov et al. 2004).

**2.** *P.7, L. 201: "Only data outgoing from the instrument upward until a relative error of 100% is reached are used, [...]". I am not sure to understand this sentence, it means you compute the relative error with respect to the radiosounding measurement and when it reaches an altitude where it is greater than 100% you take this altitude as the maximum measurement height for this period? Do you think the horizontal displacement of the radiosounding could significantly contribute to this relative error (and maybe also to the calibration)?*

Reply:
No, we did not take the radiosounding into account to get the relative error. The radiosoundings were used to get the calibration coefficient for the lidar measurements as described in Di Girolamo et al. (2016). We calculated the relative error using the absolute error of the humidity measurements and humidity measurements themselves.

**3.** *P.9, L.250: did you consider using the modification of this condition proposed by Binder2020 et al. 2020 (i.e. consider also trajectories that fulfil the 600 hPa ascent before 48h, but then descend)? Do you think that could impact the result of this study? Maybe your WCB selection was performed before the study of Binder et al. 2020, in which case I totally understand that it was not used.*

Reply:
For general WCBs we used the old definition as in Madonna et al. (2014) where the pressure difference is calculated between the maximum pressure and the pressure at the end of the trajectory length 48 h after start. For the WCB trajectories from the flights, however, we used the difference between the maximum and minimum pressure within a 48 h time interval. As in Binder et al. (2020), the latter method allows to keep WCB trajectories that did already descend from their minimum pressure before time 48 h and their pressure at time 48 h is already too high for the WCB's required pressure difference.
In our case we believe that it would not make a difference which criterion we apply because we neither see a very fast convective ascent nor any descent of the WCB in the last part of the 48 h interval.

**4.** *P.10, L.294: "On a smaller scale, the strong westerly wind ahead of the cold front initiates a secondary lee cyclone [...]". Which figure and region are you referring to with this westerly wind? Do you mean the westerly wind over Spain in Fig. 2c? In this case the Piemont region is not in the lee of this flow. Or you mean the westerly wind over France in Fig. 2a or 3b? In this case it is not ahead of the cold front.*

Reply:
We refer to Fig. 3b because the lee cyclone is fully developed there. The strong westerly wind causing the lee cyclone is best seen at 18 UTC 14 October (Fig. R6) which is not shown in the manuscript.

[Figure]

**Fig. R6:** Synoptic situation at 18:00 UTC 14 October. EDA mean equivalent potential temperature at 850 hPa (colours, K), SLP (black contours every 1 hPa), wind arrows at 850 hPa (black) and PV at 315 K (white line of 1.5 and 2 pvu). The red 'L' marks the location of the surface cyclone and the red asterisks the locations of the measurement stations.

To avoid confusion, we write now: "On a smaller scale, after 15 UTC 14 October a secondary lee cyclone occurs east of the Maritime Alps in the Italian Piedmont region (red 'L2' for the mature lee cyclone in Fig. 3b)"

**5.** *P.15, L.363: "[...] facilitates aggregation and riming[...]". Supercooled liquid water favours riming, but not necessarily aggregation. It was the wind shear and turbulence below the WCB air masses, which promoted aggregation in Gehring et al. 2020.*

Reply:
Thank you for the rectification. We corrected the text and write now: "As found by radar observations in Gehring et al. (2020), the formation of supercooled liquid water in the phase of strongest WCB ascent facilitates riming which, together with aggregation due to strong wind shear and turbulence below the WCB, would provide ideal conditions for rapid precipitation growth."

**6.** *P.15, L. 384: "[...] is likely to sublimate or evaporate." Do you have an idea why sublimation is not simulated in ERA5?*

Reply:

You mean in the EDA, right? Sublimation is a process that is implemented in the model, but, it might be underestimated. The underestimated sublimation could be related to a wrong size and fall velocity of the sedimenting particles, and/or due to errors in low-level humidity.

**7.** *P.18, L.432: "[...] if the model is assumed to underestimate precipitation from the WCB [...]". Does the model underestimate precipitation from the WCB?*

Reply:
This would be interesting to see, but the statement must stay as a hypothesis. Because precipitation is a prognostic variable we do not have it in our analysis data set.

**8.** *P.19, L.461: "The moist bias in EDA does not seem to depend on the large-scale flow direction." Do you have an hypothesis on the reason for this moist bias in EDA?*

Reply:
The study of Schäfler et al. (2011) investigated the moist bias in a WCB inflow thoroughly. They hypothesised that inaccurately represented boundary layer processes such as evapotranspiration and horizontal or vertical turbulent moisture transport might be the reason. Because we are not experts on these aspects, we would leave further analyses to the model experts.

**9.** *P.21, L.490: "[...] dynamically-driven ascent for T1 [...]". Do you know if the orography also significantly influenced the ascent of T1?*

Reply:
From Fig. 7a we would assume that the ascent of T1 is not influenced by orography as it occurs far away from orography. In contrast, T2 follows the shape of the orography for some hours very tightly while ascending. Nevertheless, this is only a subjective assessment and it would be nice to further quantify the different contributions to the ascent in further studies of WCBs near orography.

**10.** *P. 21, L.482: "[...] the orographic lifting coincided most likely with enhanced dynamical forcing for ascent [...]". The respective contribution of orographic and dynamic lifting in the ascent of the WCB is very interesting. Out of curiosity: do you think it is possible to disentangle both effects to quantify what is the contribution from each? If yes, how would you suggest to do it?*

Reply:
There is indeed a method by Demirdjian et al. (2020) that distinguishes between orographic and dynamic precipitation forcing for atmospheric rivers. They regard dynamic forcing as caused by moisture convergence while orographic forcing is indicated by moisture transport. It would be interesting to apply such a diagnostic to WCBs in future studies.

**11.** *P.24, L.430-431: "Also, a technical issue concerning the manual time adjustment of the device cannot be completely excluded." From what you wrote on P.8, L.232-233 it seems that this issue was almost confirmed. Here it sounds as if it would*

*be a possible issue that might have taken place. If this issue is likely as you mention on P.8, maybe it would be worth rephrasing this sentence and referring to P.8 (e.g. a technical issue [...] likely occurred as mentioned in Sect. 2.4). This would make it clear to the reader that it is an issue you likely had and not only a potential problem that you list among other possibilities explaining the time shift.*

Reply:
You are right, the statements are not very consistent in terms of language. The time issue is the most likely reason that came to our mind when we discussed the inconsistencies of tracer gas measurements and model-based dispersion; though we cannot be certain about this. We change the text in subsection 2.4 and express it with less certainty: ".. a technical issue concerning the manual time adjustment of the device may have occurred..."

**12.** *P.27, L.617: "[...] with a more complete instrumental package [...]". If you could redo this campaign, which instruments would you add?*

Reply:
When using in-situ instruments you must exactly target the region that you want to measure – and trust the forecast the flight planning is done with. Hence, depending on the spatial scale and shape of the targeted system, the measurements can easily fail. Remote sensing instruments in turn provide more flexibility because they cover a vertical layer. If we could do the measurements again it would be nice to get curtains from a water vapour lidar and a cloud radar.

**Technical corrections**

**13.** *P.2, L.48: "[...] an unique airborne tracer [...]". Remove the "n" in "an".*

Reply:
Thank you, done.

**14.** *P.2, L. 84: "Oertel et al. (in preparation)". I think this article is in review in WCD discussion, so it would be better to cite it as such.*

Reply:
In the meantime, this paper has been accepted. We have updated the reference.

**15.** *P.3, L.87: "This brief summary reveals the importance of WCBs for understanding precipitation in and the dynamics of extratropical cyclones, [...]". Remove the "in".*

Reply:
We prefer to keep the "in" because it emphasises the WCB as the strongest precipitation causing part of extratropical cyclones.

**16.** *P. 10, L.284: "Within the westerly flow, WCB trajectories originating over the Atlantic are embedded that start to ascend prior to passing the lidar (Fig. 2b)." I do not understand this sentence, do you mean "[...] and start to ascend [...]"?*

Reply:
Thank you, we agree with your suggestion.

**17.** *P.15, L.366 "[...] (see blue line in Fig. 3a) [...]". You probably mean Fig. 3b.*

Reply:
Thank you, Fig. 3b is correct.

**18.** *P.18, L.428 "[...] (Fig. 7a, between 16:00-19:00 UTC)." You probably mean Fig. 8a.*

Reply:
Thank you for finding the mistake, corrected.

**19.** *P.19, L.475: "[...] (read and blue dots in Fig. 8c)". It seems that there are only asterisks on Fig. 8b,c, instead of asterisk for T2 and dots for WCB or non-WCB air. This makes it hard to read when you are referring to a single asterisk in the location of T2.*

Reply:
You are right, this is confusing. There are only red and blue asterisks in Fig. 8b,c and now T1/T2 intersections are shown with a red dot. We correct the sentence.

**20.** *P.21, L. 485: "[...] when the aircraft enters the region of high WCB probabilities (Figs. 5)". It should be Fig. 5 without "s".*

Reply:
Thank you, corrected.

**21.** *P.22, caption Fig. 8: add "N" after "46.15".*

Reply:
Thank you, done.

**22.** *P.22, Fig. 8c: the labels of the WCB probability are not very clear.*

Reply:
We have improved the black lines in Fig. 8b,c.

**23.** *P.25, L. 564: "[...] (ii) if a suitable [...]". Remove "if".*

Reply:
Thank you, "if" is removed.

References:

Demirdjian, R., J. D. Doyle, C. A. Reynolds, J. R. Norris, A. C. Michaelis, and F. M. Ralph, 2020a: A Case Study of the Physical Processes Associated with the Atmospheric River Initial-Condition Sensitivity from an Adjoint Model. *J. Atmos. Sci.,* 77, 691–709, doi: 10.1175/JAS-D-19-0155.1

Khvorostyanov, V. I., and Curry, J. A., 2004: The Theory of Ice Nucleation by Heterogeneous Freezing of Deliquescent Mixed CCN. Part I: Critical Radius, Energy, and Nucleation Rate, *J. Atmos. Sci.*, *61*(22), 2676-2691, doi: 10.1175/JAS3266.1